# PHYSICS-INFORMED RESIDUAL FLOWS

## ABSTRACT

Physics-Informed Neural Networks (PINNs) embed physical laws into deep learning models. However, conventional PINNs often suffer from failure modes leading to inaccurate solutions. We trace these failure modes to two structural pathologies: gradient shattering, where gradients degrade with depth and provide little training signal, and flow mismatch, where training pushes predictions along trajectories that diverge from the PDE solution path. We introduce ResPINNs, which reformulate PINNs as residual flows, networks that iteratively refine their own predictions through explicit corrective steps, in the spirit of classical iterative solvers. Our analysis shows that this design mitigates both pathologies by keeping updates aligned with descent and by preserving informative gradients across depth. Extensive experiments on PDE benchmarks confirm that ResPINNs achieve higher accuracy with substantially fewer parameters than conventional architectures.

## 1 REVISITING FAILURE MODES IN PINNS

Partial differential equations (PDEs) govern a wide range of physical, engineering, and scientific systems. Because closed-form solutions are rarely available, numerical solvers such as finite difference, finite element, or spectral methods are the standard tools, but these approaches are computationally costly and restricted to discretized meshes. Physics-Informed Neural Networks (PINNs) (Raissi et al., 2019) have emerged as a promising alternative, embedding PDE, initial, and boundary conditions into the loss of a neural network. By leveraging automatic differentiation, PINNs can in principle approximate PDE solutions continuously in space and time.

Despite recent advances, PINNs remain vulnerable to intrinsic failure modes. Krishnapriyan et al. (2021) document several types of PDEs that are especially challenging, often due to parameters that induce high-frequency or complex solution behaviors. In such cases, PINNs may fail to propagate initial conditions accurately. A typical manifestation is the emergence of overly smooth solutions which can minimize empirical loss while ignoring temporal dynamics. To address these shortcomings, various strategies have been proposed, including optimization techniques (Wu et al., 2024; Wang et al., 2021; Liu et al., 2025; Wong et al., 2022; Bu & Karpatne, 2021), adaptive sampling (Daw et al., 2023), architectural modifications such as sequence-based models (Zhao et al., 2024; Xu et al., 2025a; Wang et al., 2021) and (Wang et al., 2024). However, these methods do not explicitly address the instability caused by noisy gradients, often referred to as gradient shattering, which can mislead training and limit robustness.

Orthogonal to these advances, we revisit the failure modes of PINNs from two complementary angles: optimization dynamics and representation flow. We identify two structural problems. First, *gradient shattering*: As depth increases, the input–output Jacobians of PINNs decorrelate exponentially, while their norms either vanish or explode. Since PDE residuals require repeated differentiation of the network outputs, this effect is amplified in PINNs, making optimization unstable even when the residual loss is small. Second, *flow mismatch*: training updates in latent space do not align with true descent directions, so the network can satisfy residual constraints locally while drifting globally, failing to propagate initial conditions. To address these issues, we introduce the notion of *residual flows*: networks designed as iterative refinement schemes, where its components perform a small correction around the identity. This stepwise view connects directly to three established perspectives: (i) residual networks, where skip connections stabilize gradients; (ii) neural ODEs, where depth corresponds to integrating a continuous-time flow; and (iii) classical iterative solvers, where predictor–corrector updates progressively reduce error. In the PINN setting, these

Table 1: Ablation study on Convection, Reaction and Wave PDEs. The relative MAE values are reported. The "–" symbol indicates a removed component and "+" indicates a replacement. For example, *PINNsFormer –Attention+MLP* replaces the attention block with an MLP. Removing attention and replacing it with linear mappings preserves or even improves performance, despite the drastic reduction in complexity.

| Model | Convection (rMAE) | Reaction (rMAE) | Wave (rMAE) |
|---|---|---|---|
| PINNsFormer (Original) | 0.510 | 0.015 | 0.270 |
| Encoder Only | 0.043 | 0.017 | 0.058 |
| -Attention + Linear | 0.012 | 0.022 | 0.022 |
| -Attention + MLP | 0.009 | 0.016 | 0.142 |
| PINNMamba (Original) | 0.019 | 0.010 | 0.020 |
| -SSM | 0.012 | 0.013 | 0.029 |
| -SSM+MLP | 0.063 | 0.014 | 0.015 |

formulations coincide: residual flows stabilize Jacobians, keep updates aligned with loss descent, and preserve initial and boundary conditions across depth.

Our paper makes the following contributions:

1. **Diagnosis.** We analyze why standard PINNs fail, tracing condition-propagation errors to *gradient shattering* and *flow mismatch*.

2. **Reformulation.** We propose *Residual Flows*, which view solution learning as stepwise refinement via small residual corrections around the identity, aligning PINNs with classical predictor–corrector methods.

3. **Evidence.** Through theory and parameter-matched ablations, we show that residual pathways—not explicit sequence modules—are the stabilizing mechanism behind recent gains. Empirical results on convection, reaction, and wave PDEs confirm improved condition preservation and solution fidelity.

## 2 DISSECTING EXISTING APPROACHES TO FAILURE MODES

A central challenge in PINNs for time-dependent PDEs is the propagation of initial conditions across time. Several recent works have sought to address this by introducing explicit sequence modeling. For example, Krishnapriyan et al. (2021) proposed recursive sequence-to-sequence training, rolling solutions forward in time with separate networks. While effective for short horizons, this strategy is memory- and compute-intensive, and does not generalize reliably outside the training window. More recent approaches have adapted modern sequence architectures: Zhao et al. (2024) introduced a transformer-based framework (PINNsFormer), while Xu et al. (2025a) proposed state-space models (PINNMamba). Both report improved accuracy and robustness, attributing their gains to the ability of attention or structured recurrence to capture long-range temporal dependencies.

At first glance, these results seem to suggest that sophisticated sequence modules are essential for overcoming failure modes in time-dependent PINNs. Yet this conclusion is not entirely satisfying: improvements could equally stem from side effects such as increased parameterization, altered optimization dynamics, or more flexible local mappings. In other words, what appears as a benefit of "long-range temporal modeling" may instead be an artifact of broader architectural changes. This motivates a sharper question: *are sequence modules truly the driving factor behind the reported improvements, or are we attributing gains to the wrong mechanism?* To probe this, we designed a controlled ablation study. Training setup, initialization, and sampling were kept fixed, and only the internal sequence modules were varied. Specifically, self-attention and state-space operators were replaced with deliberately simple local mappings (a linear projection or a shallow MLP), with parameter counts carefully matched within $\pm 10\%$. This isolates the effect of explicit sequence modeling from confounding factors such as model capacity or optimization differences.

The evidence in Table 1 challenges the conventional explanation. The table compares both transformer-based (PINNsFormer) and state-space–based (PINNMamba) architectures against ablated versions where their sequence modules are removed or replaced with simpler alternatives. For PINNsFormer, the encoder is retained while attention is stripped out and substituted either with a linear projection or a shallow MLP. For PINNMamba, the state-space operator is removed outright or replaced by an MLP with matched parameter count. Across all three PDE benchmarks, these simplified variants perform comparably to the original models, indicating that explicit attention or structured recurrence is not essential for maintaining accuracy. This suggests that the improvements attributed to sophisticated sequence modules may instead arise from a different architectural mechanism.

Across both transformer- and state-space–based PINNs, one component remains consistent: the use of *residual pathways* that carry predictions forward through incremental corrections. Unlike attention or structured recurrence, these pathways are present in every variant tested, including the simplified ablations. This observation points to residual connections—not sequence modules—as the common mechanism underlying stability and accuracy.

Why might residual pathways play such a critical role? At a high level, they enforce an update rule that keeps each layer close to the identity, nudging predictions forward through small, controlled steps rather than drastic transformations. This structure has several consequences that help explain the observed robustness:

(H1) Because updates are incremental, optimization becomes more stable: each layer only needs to make small corrections, reducing the risk of divergence.

(H2) The skip connections implicit in residual design bias the layer Jacobians toward the identity, which mitigates gradient shattering and helps preserve information across depth.

(H3) The repeated corrections accumulate like iterations of a solver, progressively refining the solution in the manner of predictor–corrector schemes.

Taken together, these hypotheses recast the source of robustness in time-dependent PINNs: not the sophistication of sequence modules, but the refinement dynamics induced by residual flows. In the remainder of this paper, we put these hypotheses to the test.

## 3 MITIGATING FAILURE MODES WITH RESIDUAL ALIGNMENT

PINNs can achieve low training loss yet still produce drifting solutions. We trace this to two mechanisms: *gradient shattering*, where Jacobians lose coherence and their norms vanish or explode with depth, degrading the derivative signal that PINNs rely on; and *flow mismatch*, where training pushes predictions along trajectories that diverge from the PDE solution. To address these issues, we view training not as a single mapping but as an *evolving flow in latent space*, advanced step by step through small residual updates. This perspective makes explicit two stabilizing principles: (i) alignment of updates with descent directions, and (ii) near-identity Jacobians that preserve gradient propagation. We begin by analyzing gradient shattering and then show how residual formulations encourage alignment.

### 3.1 PRELIMINARIES

We consider PDEs on a spatio–temporal domain $\Omega \times [0, T]$ with solution $u : \Omega \times [0, T] \to \mathbb{R}^m$ subject to interior, initial, and boundary operators $\mathcal{F}, \mathcal{I}, \mathcal{B}$:

$$\mathcal{F}(u)(x,t) = 0, \quad \mathcal{I}(u)(x,0) = 0, \quad \mathcal{B}(u)(x,t) = 0, \tag{1}$$

PINNs (Raissi et al., 2019) approximate $u$ by a neural network $u_\theta$ and train by minimizing residuals at collocation points: interior $\chi \subset \Omega \times (0, T]$, initial $\chi_0 \subset \Omega \times \{0\}$, and boundary $\chi_\partial \subset \partial\Omega \times [0, T]$. The objective is a weighted mean–squared residual,

$$L(u_\theta) = \frac{\lambda_\mathcal{F}}{|\chi|} \sum_{(x,t)\in\chi} \|\mathcal{F}(u_\theta)(x,t)\|^2 + \frac{\lambda_\mathcal{I}}{|\chi_0|} \sum_{(x,0)\in\chi_0} \|\mathcal{I}(u_\theta)(x,0)\|^2 + \frac{\lambda_\mathcal{B}}{|\chi_\partial|} \sum_{(x,t)\in\chi_\partial} \|\mathcal{B}(u_\theta)(x,t)\|^2,$$

where $\lambda_{\mathcal{F}}, \lambda_{\mathcal{I}}, \lambda_{\mathcal{B}} \geq 0$ balance the constraints.[1]

## 3.2 GRADIENT MISALIGNMENT IN PINNS

A well-documented pathology in deep networks is *gradient shattering*: correlations between input–output sensitivities at nearby inputs decay exponentially with depth, while their norms either vanish or explode depending on initialization scaling (Balduzzi et al., 2017; Poole et al., 2016; Pennington et al., 2018; Yang & Schoenholz, 2017). Since PINNs embed PDE residuals into the training objective only at sparse collocation points, they are especially vulnerable to this effect: low residuals can coexist with large solution drift between points. To formalize this, let $J_\theta(x,t) = \nabla_{(x,t)} u_\theta(x,t) \in \mathbb{R}^{m \times (d+1)}$ denote the input–output Jacobian of the network. We summarize the mean-field behavior below.

**Theorem 3.1** (Informal; mean-field gradient shattering). *Let $u_\theta$ be a depth-$L$, width-$n$ fully connected PINN with i.i.d. Gaussian initialization, a 1-Lipschitz activation and network parameters $\theta$. Denote its Jacobian $J_\theta(z) = \nabla_z u_\theta(z)$ at input $z = (x,t)$. For nearby $z'$ with $\|z' - z\| \leq r_0$, define the Frobenius cosine $\cos(J_1, J_2) = \langle J_1, J_2 \rangle_F / (\|J_1\|_F \|J_2\|_F)$. In the mean-field limit $n \to \infty$:*

   *(A) (Exponential decorrelation) $\mathbb{E}[\cos(J_\theta(z), J_\theta(z'))] = \mathcal{O}(\rho^L)$ for some $\rho \in (0, 1)$.*

   *(B) (Norm growth/decay) $\mathbb{E}\|J_\theta(z)\|_F^2 = \Theta(\gamma^L)$ for some $\gamma > 0$, with $\gamma = 1$ only at critical variance.*

*Thus, unless tuned to the edge of chaos, Jacobians decorrelate exponentially and their norms vanish or explode with depth.*

A detailed statement and proof, adapted from classical mean-field analyses of deep random networks, is provided in Appendix C.

**Implications for PINNs.** Sparse collocation makes PINNs particularly vulnerable to gradient shattering: while residuals may vanish at training points, exponential loss of Jacobian correlation and unstable norms (Theorem 3.1) allow the learned solution to drift in between. This motivates enforcing near-identity Jacobians and residual alignment mechanisms to stabilize training. Yet gradient shattering alone only explains how depth degrades the derivative signal; it does not address how individual network updates contribute to optimization. To examine this, we turn to *flow mismatch*, focusing on whether layerwise transformations align with descent directions.

## 3.3 FLOW MISMATCH CAN HURT PINNS

To understand the gradient misalignment associated with PINNs, we intrepret training as a *latent-space flow problem* indexed by an auxiliary *solver time* $k$:

$$\frac{dz(k)}{dk} = T(z(k), k; x, t), \quad k \in [0, K], \qquad z(0) = E(x, t), \tag{2}$$

where $z(k) \in \mathbb{R}^{d_h}$ is a latent state obtained from the encoding $E(x,t)$, and $T : \mathbb{R}^{d_h} \to \mathbb{R}^{d_h}$ denotes the residual transformation that advances $z(k)$ toward the PDE solution. This operator may be fixed (as in classical solvers) or learned (as in neural architectures introduced later). In discrete form,

$$z_{k+1} = z_k + T_k(z_k; \alpha), \qquad k = 0, \dots, K-1,$$

with step parameter $\alpha > 0$ implicit in $T_k$. When $\|T_k\|$ is small, each update is a residual correction around the identity. This lens makes two optimization effects explicit: (i) *iterative refinement*, where many small, well-aligned corrections reduce the loss predictably; and (ii) *Jacobian neutrality*, where near-identity Jacobians stabilize gradient propagation across depth. Consider a composition of $K$ such transformations and a loss function $\mathcal{L}(z_k)$ on the $k^{th}$ transformation .

**Lemma 3.2** (Local update descent with depth-aware smoothness). *Let $z_{k+1} = z_k + T_k(z_k)$ with Jacobian $J_k := \partial z_{k+1} / \partial z_k = I + A_k$. If $\mathcal{L}$ has $\beta$-Lipschitz continous gradient in a neighborhood*

---

[1]We use $\langle A, B \rangle_F = \text{tr}(A^\top B)$, $\|A\|_F$ for Frobenius norms, $\| \cdot \|$ for Euclidean norms, and $\| \cdot \|_2$ for spectral norm

*of $z_k$, then there exist*

$$\beta_k \ \leq \ \beta \Big( \prod_{\ell=k}^{K-1} \|J_\ell\|_2 \Big)^2$$

*such that* $\mathcal{L}(z_{k+1}) \ \leq \ \mathcal{L}(z_k) \ + \ \langle \nabla_{z_k}\mathcal{L}(z_k)^\top, T_k \rangle + \ \frac{\beta_k}{2}\|T_k\|^2.$

This follows from a first-order Taylor expansion of the $\mathcal{L}(z_{k+1}) := \mathcal{L}(z_k + T(z_k))$. The formal statement and proofs are deferred to the Appendix B. Notice that a first order term is a good approximation when the magnitude of $T_k$ is small enough. Rolling out Lemma 3.2 over $k = 0, \ldots, K-1$ updates recursively gives

$$\mathcal{L}(z_K) \ \leq \ \mathcal{L}(z_0) \ + \ \sum_{k=0}^{K-1}\Big[\langle \nabla_{z_k}\mathcal{L}(z_k)^\top, T_k \rangle + \ \frac{\beta_k}{2}\|T_k\|^2\Big].$$

In particular, Lemma 3.2 implies that the first-order change in the loss at step $k$ is driven by the dot product between the local loss gradient and $T_k$,

$$\mathcal{L}(z_{k+1}) - \mathcal{L}(z_k) \ \approx \ \langle \nabla_{z_k}\mathcal{L}(z_K), T_k(z_k)\rangle.$$

We can characterize this via *gradient alignment*: the cosine between the step and the negative gradient,

$$\text{GA}_k \ := \ \frac{\langle T_k(z_k), -\nabla_{z_k}\mathcal{L}(z_k)\rangle}{\|T_k(z_k)\|\,\|\nabla_{z_k}\mathcal{L}(z_k)\|},$$

so that

$$\langle \nabla_{z_k}\mathcal{L}(z_k), T_k(z_k)\rangle = \ -\|T_k(z_k)\|\,\|\nabla_{z_k}\mathcal{L}(z_k)\|\,\text{GA}_k.$$

Thus, each local update constitutes a gradient-based step whose contribution is exactly proportional to its alignment with $-\nabla_{z_k}\mathcal{L}(z_k)$: $\text{GA}_k > 0$ moves $z_k$ into the descent half-space (first-order decrease), $\text{GA}_k = 0$ is neutral, and $\text{GA}_k < 0$ moves uphill. Stable descent therefore requires residual updates to remain small and aligned with the loss gradient. *Flow mismatch* denotes the opposite regime, where updates are too large or alignment is negative, causing predictions to drift despite decreasing loss.

However, alignment alone is not sufficient: even well-aligned updates can suffer from vanishing or exploding gradients if Jacobian spectra are uncontrolled. As a complementary effect, residual formulations also encourage near-identity Jacobians, as shown next.

**Theorem 3.3** (Local Jacobian Neutrality). *If $J_k = I + A_k$ with $\|A_k\|_2 \leq \alpha_k < 1$, then*

$$1 - \alpha_k \leq \sigma_{\min}(J_k) \leq \sigma_{\max}(J_k) \leq 1 + \alpha_k, \qquad \kappa(J_k) \leq \frac{1 + \alpha_k}{1 - \alpha_k},$$

*and for all $v$, $(1 - \alpha_k)\|v\| \leq \|J_k^\top v\| \leq (1 + \alpha_k)\|v\|$. $\alpha_k$ denotes an upper bound on the spectral norm of the deviation from identity at step $k$.*

*Proof sketch.* Weyl's inequality gives $|\sigma_i(J_k) - 1| \leq \|A_k\|_2$; the bounds follow immediately.

Thus, *if* residual updates are small (small $\alpha_k$), per-step Jacobians remain near identity, stabilizing gradient propagation across depth. Residual flow formulations exhibit the three predicted properties: they maintain positive gradient alignment, have near-identity Jacobians, and operate in the small-step regime. Figure 1 (and Appendix 8) illustrates this empirically. This aligns with earlier observations that residual connections implement iterative inference (Greff et al., 2017; Jastrzebski et al., 2018).

*Appendix roadmap.* Appendix B: proofs of Lemma 3.2 and Theorem 3.3, plus guarantees for residual flows. Appendix C: note on gradient shattering.

## 4 RELATED WORK

**Neural Operators.** Neural operators such as DeepONet (Lu et al., 2021a) and FNOs (Li et al., 2023) are data driven *surrogate models* that approaximate the PDE solution operator from labeled data, whereas PINNs rely on enforcing PDE residuals and boundary/initial conditions. Since our work focuses on PINNs, we benchmark mainly against PINN variants and restrict our analysis to failure modes specific to this class of methods.

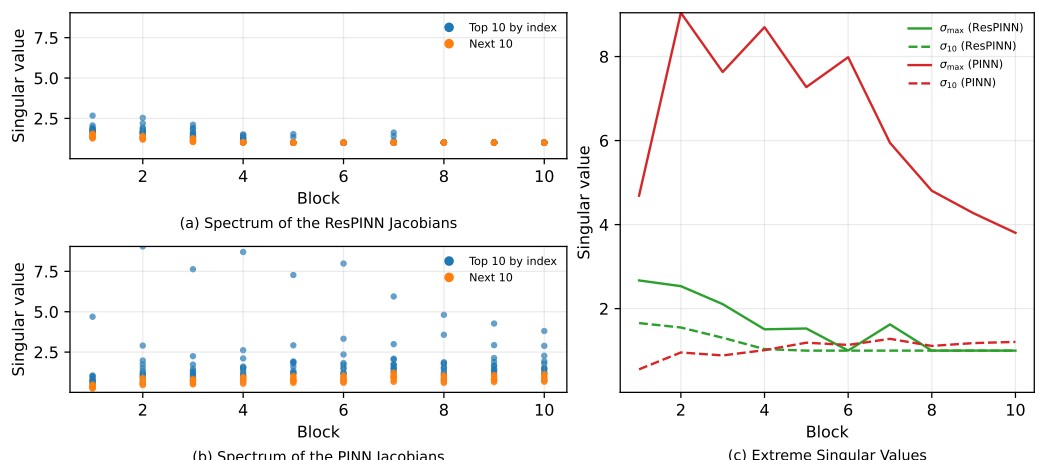

Figure 1: Spectral analysis of Jacobians across network depth for the 1D convection problem. (a) ResPINN (with residual connections): singular values remain clustered near unity, indicating near-isometric mappings and stable gradients. (b) Standard PINN: singular values remain spread with depth, reflecting anisotropy and poor conditioning. (c) Extreme singular values ($\sigma_{\max}$ and $\sigma_{10}$) highlight the contrast: PINNs amplify dominant directions, whereas ResPINNs suppress spectral growth. Both architectures use the same number of parameters; the only difference is the residual connection inserted after every two layers.

**Flows in Machine Learning.** The idea of flows is well-established in Machine Learning. Normalizing flows learn invertible maps that transform a base density into a data density by composing simple bijections and tracking Jacobian determinants for exact likelihoods (Rezende & Mohamed, 2015; Dinh et al., 2017; Kingma & Dhariwal, 2018; Chen et al., 2018). This probabilistic goal is orthogonal to ours: we do not model densities or require invertibility. Our flow perspective instead concerns optimization dynamics of PINNs exhibiting failure modes.

**Residual networks and connections.** Residual connections stabilize training by composing near-identity transformations (He et al., 2016; Greff et al., 2017; Jastrzebski et al., 2018; Haber & Ruthotto, 2017; Lu et al., 2018; Chen et al., 2018). In PINNs, they appear in several proposals, often alongside other architectural changes, so their specific contribution is unclear (Wang et al., 2020; Zhao et al., 2024; Xu et al., 2025a). PINNsFormer (Zhao et al., 2024) and PINNMamba (Xu et al., 2025a) both employ residual connections but attribute gains to sequence modeling, while PirateNets (Wang et al., 2024) explore adaptive residual scaling and physics-informed initialization without connecting them to failure modes.

**Continuous-depth limits and Neural ODEs.** Taking residual networks to the continuous-depth limit yields Neural ODEs, parameterized by vector fields and solved numerically (Chen et al., 2018). Connections to solver stability and residual architectures have been emphasized (Lu et al., 2018; Haber et al., 2019), and continuous-depth models have been adapted to scientific machine learning (Yin et al., 2023; Verma et al., 2024).

**Positioning.** Our contribution differs from likelihood-based flows and prior PINN adaptations. We explicitly characterize two structural failure modes in PINNs, gradient shattering and flow mismatch, and propose a residual flow formulation that enforces gradient alignment and Jacobian neutrality. The mechanism is architecture-agnostic: it can be instantiated as a residual stack, as a continuous-depth Neural ODE with explicit solvers, or as a purely iterative refinement scheme without ODE machinery. This unifies discrete residual nets, continuous flows, and solver-style iterations under a single stabilization principle tailored to PINNs.

## 5 EMPIRICAL EVALUATION

**Architectures.** Our proposal is to view PINNs through the lens of *residual flows*: neural networks that refine predictions iteratively, analogous to numerical solvers advancing a state over time. To investigate this perspective, we consider three architectural variants that will serve as the basis of our analysis (described below):

- **Residual Networks (ResPINN).** Discrete residual flows, where each block applies a correction around the identity $h_{k+1} = h_k + \alpha f(h_k; \theta_k)$ with $h_k$ denoting the hidden state (block input/output) at depth $k$, and $f(\cdot; \theta_k)$ the learned transformation within that block..

- **Neural ODEs (O-PINN).** Continuous residual flows obtained in the infinitesimal-step limit, integrating $\dot{h} = f_\theta(h, t)$ with a numerical ODE solver (Chen et al., 2018). This provides the continuous-depth analogue of the residual formulation.

- **Progressive Residual Flows.** A curriculum-style variant of residual networks that increases depth gradually during training by appending new residual blocks while freezing earlier ones. This mirrors multistage solvers where successive corrections extend accuracy.

**Benchmarks.** We evaluate on four established benchmarks. First, three canonical time-dependent PDEs—Wave, Reaction, and Convection—are widely used to probe optimization behavior in PINNs (Raissi et al., 2019; Krishnapriyan et al., 2021; Zhao et al., 2024; Wu et al., 2024) and Heat Equation. Prior work has shown that Reaction–Diffusion and Convection in particular expose common failure modes of PINNs (Krishnapriyan et al., 2021). Moreover, we include the *PINNacle* suite (Zhongkai et al., 2024), a collection of 16 diverse PDE tasks spanning Burgers, Poisson, Heat, Navier–Stokes, Wave. Detailed formulations, discretizations, and training domains are given in Appendix D.

**Baselines.** We compare against a broad suite of PINN architectures, spanning classical approaches (MLP-based PINNs (Raissi et al., 2019), FLS (Wong et al., 2022), QRes (Bu & Karpatne, 2021)), recent improvements (KANs (Liu et al., 2025), state-of-the-art sequential models (PINNsFormer (Zhao et al., 2024), PINNMamba (Xu et al., 2025a)) and two methods closely related to our residual-flow perspective: PirateNet (Wang et al., 2024), which employs adaptive residual networks, and SOAP_PINN (Wang et al., 2025), which stabilizes training via gradient-alignment–based second-order optimization. This collection includes both pointwise networks and methods explicitly designed to address failure modes in dynamical systems using sequence modeling approaches.

**Implementation.** We instantiate the latent *residual flow* architectures as block-structured networks. Unless otherwise noted, all models are trained on $101 \times 101$ space–time grids using the L–BFGS optimizer and the wavelet activation of Zhao et al. (2024). For the baselines, we follow the original configurations: *PINNMamba* uses subsequences of length 7 with step size $10^{-2}$, and *PINNsformer* uses subsequences of length 5 with step size $10^{-4}$. All other models operate without subsequencing. For the *PINNacle* benchmark, dataset sizes and sampling details are provided in Appendix E. Residual flow blocks use a hidden dimension of 64, with three fully connected layers per block followed by a skip connection. The stagewise variant begins with three blocks and adds two new blocks at each stage, freezing the earlier ones. Neural ODE variants integrate a single residual block parameterization with a 4th-Order Runge-Kutta(RK4) solver (See Appendix F).

### 5.1 DO RESIDUAL FLOWS MITIGATE FAILURE MODES?

We first benchmark *ResPINN* against recent PINN variants. Table 2 reports relative mean absolute error (rMAE) and relative root mean squared error (rRMSE) (See Appendix D for more details about the metrics). Classical PINNs perform poorly on Reaction and Convection, consistent with known failure modes. Both *PINNsFormer* and *PINNMamba* incorporate residual connections, but only at the level of one or two residual blocks. In contrast, *ResPINN* stacks residual updates throughout the network, directly instantiating the residual flow formulation. Across all four PDEs, ResPINN achieves the lowest errors, often by an order of magnitude, showing that residual flows provide consistent improvements beyond the shallow residual structures of prior models. Qualitative comparisons in Figure 2 confirm this pattern: Models with residual pathways achieve constructive reconstructions whereas vanilla PINNs suffer a larger deviation.

Table 2: Quantitative results on four PDE benchmarks. ResPINN consistently outperforms baselines.

| Model | Wave | | Reaction | | Convection | | Heat | |
|---|---|---|---|---|---|---|---|---|
| | rMAE | rRMSE | rMAE | rRMSE | rMAE | rRMSE | rMAE | rRMSE |
| PINNs | 0.4101 | 0.4141 | 0.9803 | 0.9785 | 0.8514 | 0.8989 | 0.8956 | 0.9404 |
| QRes | 0.5349 | 0.5265 | 0.9826 | 0.9830 | 0.9035 | 0.9245 | 0.8381 | 0.8800 |
| FLS | 0.1020 | 0.1190 | 0.0220 | 0.0390 | 0.1730 | 0.1970 | 0.7491 | 0.7866 |
| PINNsFormer | 0.3559 | 0.3632 | 0.0146 | 0.0296 | 0.4527 | 0.5217 | 0.2129 | 0.2236 |
| RoPINNs | 0.1650 | 0.1720 | 0.0070 | 0.0170 | 0.6350 | 0.7200 | 0.1545 | 0.1622 |
| KAN | 0.1433 | 0.1458 | 0.0166 | 0.0343 | 0.6049 | 0.6587 | 0.0901 | 0.1042 |
| PINNMamba | 0.0197 | 0.0199 | 0.0094 | 0.0217 | 0.0188 | 0.0201 | 0.0535 | 0.0583 |
| PINN_SOAP | 0.2825 | 0.2851 | 0.0048 | 0.0096 | 0.0340 | 0.0363 | 0.0098 | 0.0086 |
| PirateNet | 0.2544 | 0.2637 | 0.0589 | 0.0965 | 0.9704 | 0.9704 | **0.0005** | **0.0005** |
| **ResPINN (ours)** | **0.0130** | **0.0154** | **0.0047** | **0.0075** | **0.0028** | **0.0046** | 0.0035 | 0.0048 |

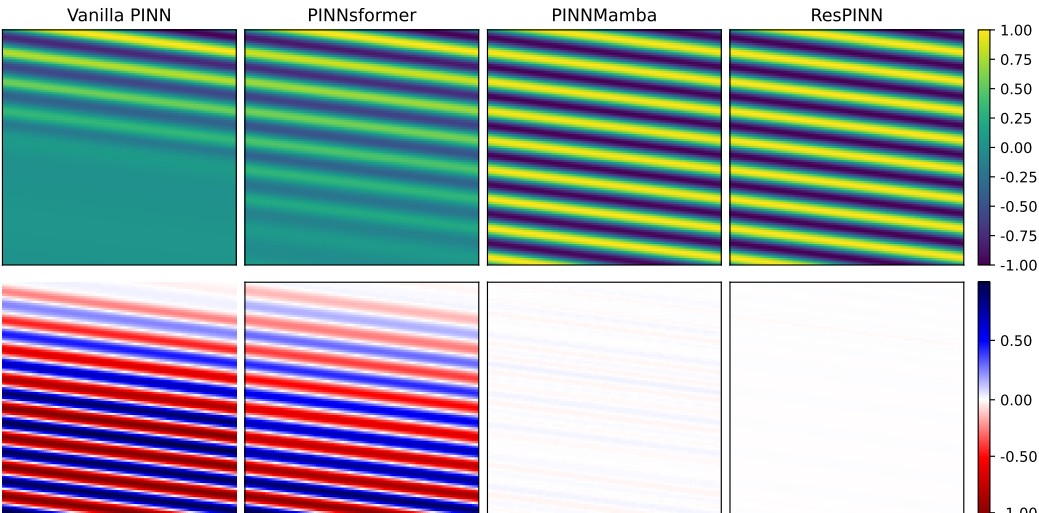

Figure 2: Qualitative comparison on Convection PDE. Top: predicted solutions. Bottom: pointwise errors.

## 5.2 ITERATIVE REFINEMENT AND GRADIENT ALIGNMENT

We next ask whether residual flows in PINNs act primarily as feature learners or as iterative refiners of predictions. We investigate this from two complementary perspectives.

For each block $T_i$, we measure the relative update size $\frac{\|T_i(z_i)\|}{\|z_i\|}$ averaged across sample points. Large values indicate substantial representation change (feature learning), while small values indicate incremental corrections (refinement). Figure 3 shows that in standard PINNs the ratio remains large across depth, whereas in ResPINNs it decreases steadily, consistent with refinement dynamics. For details on other PDEs, see Appendix H. To probe whether the individual blocks can contribute to failure modes, we adopt the progressive-flow setting. At each training stage, after adding new residual blocks and freezing earlier ones, we train only a linear projection head to read out predictions from intermediate stages. Figure 4 in Appendix H illustrates that early stages incur high error similar to failure modes, but later stages systematically reduce error while leaving earlier predictions unchanged. This confirms that new blocks act as refiners rather than relearners, mirroring multistage correction in classical solvers.

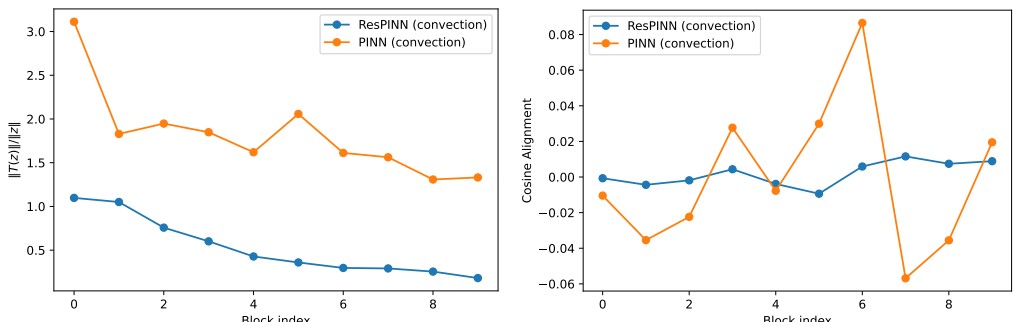

Figure 3: Left: relative update size $\|T_i(z_i)\|/\|z_i\|$ across depth. ResPINNs produce progressively smaller corrections, consistent with refinement. Right: Gradient Alignment. ResPINNs exhibit an almost neutral alignment with the local gradient descent.

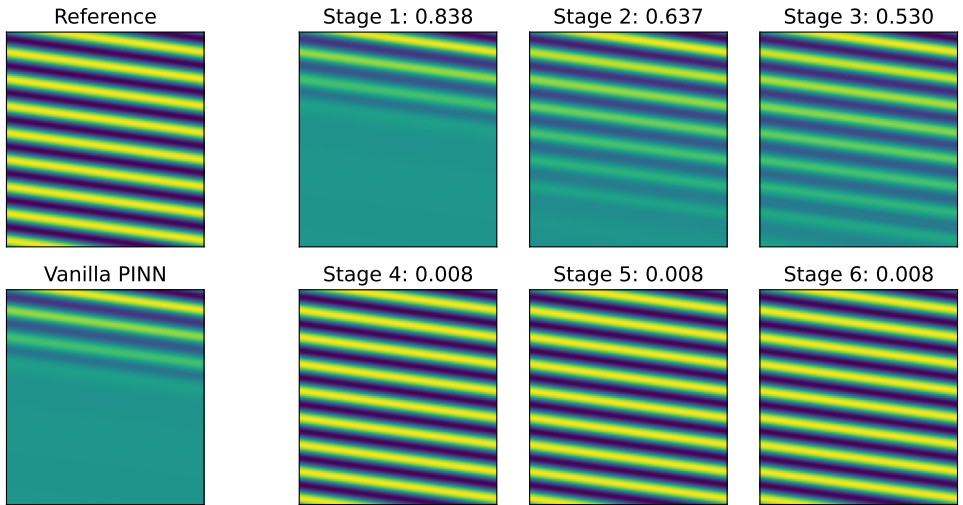

Figure 4: Predicted solutions across stages of the progressive residual flow procedure. The left column shows the reference prediction (top) and the vanilla PINN baseline (bottom) for the convection problem. The right panels present stagewise predictions annotated with stage index and rMSE. Early stages exhibit failure behavior and miss the temporal structure of the solution, while later stages progressively recover finer temporal dynamics.

## 5.3 ABLATION STUDY

To disentangle the effect of discretization from architectural or activation choices, we compare the continuous-depth formulation (*O-PINN*, integrated with a fixed-RK4 ODE solver) against its discrete counterpart (*ResPINN*), each trained with either $\texttt{tanh}$ or wavelet activations. This ablation allows us to test whether the improvements stem from the residual flow discretization itself or from particular activation functions. The results in Table 3 show that O-PINN and ResPINN exhibit complementary strengths: the continuous formulation benefits some PDE families (especially with wavelet activations), while discrete residual stacks remain competitive elsewhere.

## 5.4 EXPERIMENTS ON COMPLEX PROBLEMS

To assess generalization, we evaluate on *PINNacle* (Zhongkai et al., 2024). On challenging multi-scale tasks, baselines such as PINNsFormer (Zhao et al., 2024) and PINNMamba (Xu et al., 2025a) either fail to converge or run into out-of-memory errors, whereas *ResPINN* trains successfully while maintaining comparable accuracy on the remaining tasks. Details of the PINNacle experiments are shown in Appendix E. In addition, to further evaluate robustness on harder PDEs beyond the

| | Wave | | Reaction | | Convection | |
|---|---|---|---|---|---|---|
| Model | rMAE | rRMSE | rMAE | rRMSE | rMAE | rRMSE |
| O-PINN + $\tanh$ | 0.038 | 0.039 | 0.018 | 0.035 | 0.014 | 0.016 |
| O-PINN + wavelet | 0.053 | 0.059 | 0.003 | 0.005 | 0.003 | 0.003 |
| ResPINN + $\tanh$ | 0.030 | 0.030 | 0.008 | 0.017 | 0.015 | 0.016 |
| ResPINN + wavelet | 0.070 | 0.074 | 0.008 | 0.009 | 0.006 | 0.006 |

Table 3: Ablation on activation functions for continuous (O-PINN) and discrete (ResPINN) residual flow models. Results are reported on Wave, Reaction, and Convection PDEs using relative rMAE and rRMSE.

PINNacle suite, we include results on five additional equations: Allen–Cahn, Korteweg–de Vries, Ginzburg–Landau, lid-driven cavity, and Rayleigh–Taylor instability, comparing *ResPINN* with PirateNet (Wang et al., 2024). As reported in Table 6, ResPINN achieves comparable or lower $L^2$ error across all cases, demonstrating stable performance even under complex nonlinear and multi-scale dynamics.

## 6 CONCLUSION

We reframed PINNs as *residual flows*: networks that solve PDEs by iteratively refining predictions through small residual updates. This view makes two optimzation effects explicit—*gradient alignment* (updates aligned with descent) and *Jacobian neutrality* (near-identity per-step Jacobians)—and led to simple instantiations (ResPINN, O-PINN, progressive residual flows).

Across canonical PDEs and the PINNacle suite, ResPINN achieved consistently lower errors. Mechanistic diagnostics support our hypotheses: residual blocks operate in the small-step regime (H1), maintain near-identity Jacobians across depth (H2), and exhibit iterative refinement (H3) as shown by decreasing update ratios in ResPINNs and stagewise error telescoping in curriculum training. These gains persist across activations, and the continuous formulation (O-PINN) can be advantageous on some PDE families, suggesting that continuous-time parameterizations merit exploration for very deep regimes.

Future work may explore how different numerical solvers induce distinct refinement behaviors, and whether ODE parameterizations applied directly in solution space for time-dependent PDEs can further mitigate failure modes. Bridging local theoretical insights with global behaviors observed in practice offers a promising avenue for deepening our understanding of residual flows.

## 7 REPRODUCIBILITY STATEMENT.

All PDE setups (governing equations, domains, analytic solutions, and meshes) are detailed in Appendix D. Theoretical results and proofs appear in Appendix B, with the mean-field shattering adaptation in Appendix C. Architectural and solver specifications for ResPINN, O-PINN, and Progressive Flow are given in Appendix F, and additional alignment/refinement diagnostics are in Appendix H. PINNacle task definitions and results are reported in Appendix E. An anonymous code respository containing implementations of residual flows and scripts reproducing the experiments is available at `https://anonymous.4open.science/r/resflows-0FD5`

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

## A    THE USE OF LARGE LANGUAGE MODELS (LLMS)

We used large language models solely for surface-level editing: spelling and grammar correction, and minor wording improvements. LLMs were *not* used for idea generation, experiment design, data analysis, coding, mathematical derivations, or substantive content creation.

## B    PROOFS

We study feature evolution through a latent flow induced by residual transformations in continuous network time:

$$\frac{dz(k)}{dk} \;=\; T\big(z(k), k;\, x, t\big), \qquad k \in [0, K], \qquad z(0) = z_0 := E(x, t) \in \mathbb{R}^{d_h}. \tag{3}$$

*Remark* B.1 (Analytical surrogate). Equation equation 3 is not intended as the literal dynamics of fully connected PINNs, but as an analytical surrogate that lets us study feature evolution and gradient misalignment using the language of residual flows.

**Lemma B.2** (Integral form). *If $T(\cdot, \cdot)$ is continuous, then $z$ is a solution of equation 3 on $[0, K]$ if and only if*

$$z(k) \;=\; z_0 + \int_0^k T\big(z(\tau), \tau;\, x, t\big)\, d\tau, \qquad 0 \le k \le K. \tag{4}$$

Proof. *(⇒) Integrate equation 3 from 0 to $k$ to obtain equation 4.*
*(⇐) If equation 4 holds and $T(z(\tau), \tau)$ is continuous in $\tau$, then by the fundamental theorem of calculus the map $k \mapsto z(k)$ is differentiable with $\frac{dz}{dk} = T(z(k), k)$ and $z(0) = z_0$, i.e., $z$ solves equation 3.* □

**Theorem B.3** (Banach contraction mapping). *Let $(X, \|\cdot\|_X)$ be a Banach space and let $F : X \to X$ satisfy*
$$\|F(z) - F(z')\|_X \;\le\; c\, \|z - z'\|_X, \qquad \forall z, z' \in X,$$
*for some $0 < c < 1$. Then $F$ admits a unique fixed point $z^* \in X$, and the iterates $z^{(n+1)} = F(z^{(n)})$ converge to $z^*$ for any initial $z^{(0)} \in X$.*

**Theorem B.4** (Existence and uniqueness of a solution of a residual flow). *Let $T : \mathbb{R}^{d_h} \times [0, K] \to \mathbb{R}^{d_h}$ be continuous and assume there exists $L > 0$ such that*

$$\|T(z_1, k) - T(z_2, k)\| \;\le\; L\, \|z_1 - z_2\|, \qquad \|T(z, k)\| \;\le\; L(1 + \|z\|), \tag{5}$$

*for all $z, z_1, z_2 \in \mathbb{R}^{d_h}$ and $k \in [0, K]$. Then the IVP 3 admits a unique solution $z \in C([0, K], \mathbb{R}^{d_h})$.*

*Proof.* Fix $\delta > 0$ and consider the Banach space $X = C([0, \delta], \mathbb{R}^{d_h})$ with norm $\|z\|_X = \sup_{0 \le s \le \delta} \|z(s)\|$. Define the flow operator

$$(\mathcal{F}z)(k) \;:=\; z_0 + \int_0^k T(z(\tau), \tau)\, d\tau.$$

If $\mathcal{F}z = z$, then by Lemma B.2, $z$ solves the IVP on $[0, \delta]$.

For $z, z' \in X$ and $k \in [0, \delta]$,

$$\|(\mathcal{F}z)(k) - (\mathcal{F}z')(k)\| \le \int_0^k \|T(z(\tau), \tau) - T(z'(\tau), \tau)\|\, d\tau \le L\delta\|z - z'\|_X.$$

Thus $\|\mathcal{F}z - \mathcal{F}z'\|_X \le L\delta\|z - z'\|_X$, so $\mathcal{F}$ is a contraction whenever $L\delta < 1$. By Theorem B.3, $\mathcal{F}$ has a unique fixed point in $X$, which is the unique solution on $[0, \delta]$. Repeating the argument on successive intervals of length $\delta$ extends the solution uniquely to all of $[0, K]$. □

**Definition B.5** (Discrete Residual Step). Let $\Delta k > 0$ and $k_n := n \Delta k$ for $n = 0, \ldots, N$ with $N\Delta k = K$. The explicit Euler discretization of the residual flow $\frac{dz(k)}{dk} = T(z(k), k; x, t)$ with $z(0) = z_0 := E(x, t)$ is

$$z_{n+1} = z_n + \Delta k\, T(z_n, k_n; x, t), \qquad z_0 = E(x, t).$$

Equivalently, this is a residual update with $T_n(z_n) := \Delta k\, T(z_n, k_n; x, t)$.

**Definition B.6** (Convergence/order). Let $z(\cdot)$ denote the (unique) solution of the IVP on $[0, K]$. A time-stepping scheme producing $\{z_n\}_{n=0}^N$ is said to *converge with order $p$* on $[0, K]$ if there exists a constant $C$, independent of $\Delta k$, such that

$$\max_{0 \le n \le N} \left\| z(k_n) - z_n \right\| \le C\, (\Delta k)^p.$$

**Theorem B.7** (First-order convergence of the residual flows). *Assume the hypotheses of existence/uniqueness hold (global Lipschitz and linear growth in $z$ for $T$), and that the solution $z$ is twice continuously differentiable on $[0, K]$. Let $\{z_n\}$ be defined by B.5. Then the discrete formulation of the residual flows converges with order $1$:*

$$\max_{0 \le n \le N} \left\| z(k_n) - z_n \right\| \le C_K\, \Delta k,$$

*where $C_K$ depends on $K$, the Lipschitz constant $L$ of $T$ in $z$, and $\max_{k \in [0,K]} \|\ddot{z}(k)\|$, but is independent of $\Delta k$.* Sketch. *Taylor expand $z(k_{n+1}) = z(k_n) + \Delta k\, \dot{z}(k_n) + R_n$ with $\|R_n\| \le C\,(\Delta k)^2$. Using $\dot{z}(k_n) = T(z(k_n), k_n)$ and subtracting the Euler step gives the error recurrence $e_{n+1} \le (1 + L\Delta k)\, e_n + C\,(\Delta k)^2$, where $e_n := \|z(k_n) - z_n\|$. Apply the discrete Grönwall lemma to obtain $e_n \le C\, \frac{e^{Lk_n} - 1}{L}\, \Delta k \le C\, \frac{e^{LK} - 1}{L}\, \Delta k$.*

**Proposition B.8** (Gradient alignment in residual flows). *Let $\mathcal{L} : \mathbb{R}^{d_h} \to \mathbb{R}$ be continuously differentiable, and let $z : [0, K] \to \mathbb{R}^{d_h}$ be a continuously differentiable solution of the residual flow IVP equation 3. Then, for all $k \in [0, K]$,*

$$\frac{d}{dk}\, \mathcal{L}(z(k)) = \left\langle \nabla \mathcal{L}(z(k)),\, T(z(k), k; x, t) \right\rangle. \tag{6}$$

*1. If*

$$\left\langle \nabla \mathcal{L}(z(k)),\, T(z(k), k) \right\rangle \le 0 \quad \text{for all } k \in [0, K], \tag{7}$$

   *then $\mathcal{L}(z(k))$ is nonincreasing on $[0, K]$.*

*2. If there exists a constant $c \in (0, 1]$ such that*

$$\frac{\left\langle T(z(k), k),\, -\nabla \mathcal{L}(z(k)) \right\rangle}{\|T(z(k), k)\|\, \|\nabla \mathcal{L}(z(k))\|} \ge c \quad \text{and} \quad \|T(z(k), k)\| > 0 \quad \text{for all } k \in I \subset [0, K], \tag{8}$$

   *then $\mathcal{L}(z(k))$ is strictly decreasing on $I$.*

*Proof.* The chain rule gives equation 6. Under equation 7, $\frac{d}{dk}\mathcal{L}(z(k)) \le 0$ for all $k$, so $\mathcal{L}(z(k))$ is nonincreasing.

For equation 8, write

$$\frac{d}{dk}\, \mathcal{L}(z(k)) = -\, \|\nabla \mathcal{L}(z(k))\|\, \|T(z(k), k)\|\, \frac{\left\langle T(z(k), k),\, -\nabla \mathcal{L}(z(k)) \right\rangle}{\|T(z(k), k)\|\, \|\nabla \mathcal{L}(z(k))\|} \tag{9}$$

$$\le -c\, \|\nabla \mathcal{L}(z(k))\|\, \|T(z(k), k)\|. \tag{10}$$

On any interval $I$ where $c > 0$ and $\|T(z(k), k)\| > 0$, the right-hand side is strictly negative, hence $\mathcal{L}(z(k))$ is strictly decreasing on $I$. $\qquad\square$

*Proof of Lemma 3.2.* Since $\mathcal{L}$ has $\beta$-Lipschitz continuous gradient, we have for any $u, v$ in a neighborhood of $z_k$:

$$\mathcal{L}(v) \le \mathcal{L}(u) + \langle \nabla \mathcal{L}(u), v - u \rangle + \frac{\beta}{2} \|v - u\|^2.$$

Apply this inequality with $u = z_k$ and $v = z_{k+1} = z_k + T_k(z_k)$:

$$\mathcal{L}(z_{k+1}) \le \mathcal{L}(z_k) + \langle \nabla_{z_k} \mathcal{L}(z_k), T_k(z_k) \rangle + \frac{\beta}{2} \|T_k(z_k)\|^2.$$

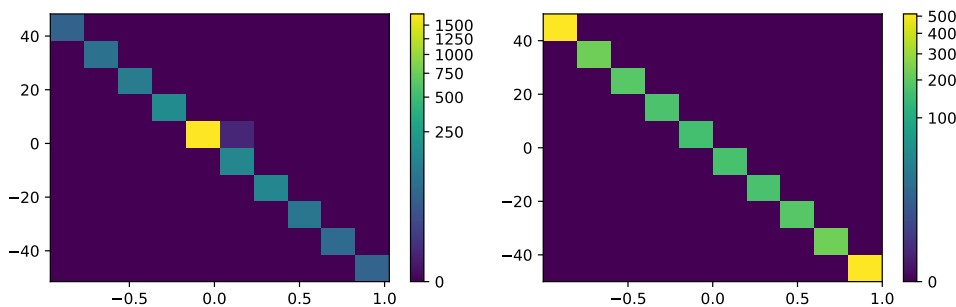

Figure 5: A 2D histogram of input-output Jacobian for convection PDE evaluated on a 50x50 equi-spaced grid. The plots show spatial (y-axis) and temporal (x-axis) PDE residuals. (left) A trained vanilla PINN. Most gradients are concentrated around 0 which is a sign of vanishing gradients. (right) The Jacobian of the analytical solution.

To capture the effect of depth, note that subsequent updates depend on how $T_j$ is transformed through the Jacobians $J_\ell = \partial z_{\ell+1}/\partial z_\ell$. The gradient at $z_k$ is related to the gradient at $z_{k+1}$ by the chain rule:

$$\nabla_{z_k}\mathcal{L}(z_{k+1}) = J_k^\top \nabla_{z_{k+1}}\mathcal{L}(z_{k+1}).$$

Rolling this back from step $K$ to step $k$ shows that each local Lipschitz constant is scaled by the squared operator norms of the Jacobians:

$$\|\nabla^2 \mathcal{L}(z_k)\|_2 \ \leq \ \beta\Big(\prod_{\ell=k}^{K-1} \|J_\ell\|_2\Big)^2.$$

Therefore there exists a local smoothness constant $\beta_k \leq \beta\big(\prod_{\ell=k}^{K-1}\|J_\ell\|_2\big)^2$ such that

$$\mathcal{L}(z_{k+1}) \leq \mathcal{L}(z_k) + \langle \nabla_{z_k}\mathcal{L}(z_k), T_k(z_k)\rangle + \frac{\beta_k}{2}\|T_k(z_k)\|^2.$$

This completes the proof. $\qquad\square$

## C    MEAN-FIELD GRADIENT SHATTERING FOR PINN JACOBIANS

Our analysis of gradient shattering follows directly from the mean-field studies of deep random networks by Poole et al. (2016), Balduzzi et al. (2017), Pennington et al. (2018), and Yang & Schoenholz (2017). We adapt their derivations to the input–output Jacobians relevant for PINNs.

**Theorem C.1** (Adapted from prior work on shattered gradients)**.** *Consider a depth-$L$, width-$n$ fully-connected network with random Gaussian initialization as in Poole et al. (2016); Balduzzi et al. (2017). Let $J_\theta(z) = \nabla_z u_\theta(z)$ denote the input–output Jacobian at input $z$. In the mean-field limit $n \to \infty$ the following hold:*

*(A) **Exponential decorrelation.** For nearby inputs $z, z'$, correlations between Jacobians decay exponentially with depth:*

$$\mathbb{E}\big[\cos(J_\theta(z), J_\theta(z'))\big] = \mathcal{O}(\rho^L), \quad \rho \in (0,1).$$

*(B) **Norm growth/decay.** Jacobian norms scale exponentially with depth:*

$$\mathbb{E}\|J_\theta(z)\|_F^2 = \Theta(\gamma^L),$$

*with $\gamma = 1$ only on the edge-of-chaos manifold; generically $\gamma \neq 1$, yielding vanishing or exploding norms.*

**Proof sketch.** The argument mirrors Poole et al. (2016); Pennington et al. (2018). Pre-activations converge to Gaussian processes in the mean-field limit, and input sensitivities evolve via multiplicative recursions depending on $\mathbb{E}[\phi'(u)^2]$. Cross-input correlations shrink by a factor $\rho < 1$ per layer, while sensitivity norms scale by $\gamma$.

*Proof.* Write the depth-$L$ fully-connected network as

$$h^0(z) = z, \qquad h^\ell(z) = W^\ell \phi(h^{\ell-1}(z)), \quad \ell = 1, \ldots, L,$$

where $W_{ij}^\ell \sim \mathcal{N}(0, \sigma_w^2/n)$ i.i.d., biases are omitted for simplicity, and $u_\theta(z) = v^\top \phi(h^L(z))$ with $v_i \sim \mathcal{N}(0, \sigma_v^2/n)$ independent of the $W^\ell$. $n$ denotes the number of neurons.

**Mean-field limit.** In the limit $n \to \infty$ the preactivations $\{h_i^\ell(z)\}_{i=1}^n$ form i.i.d. Gaussians with variance $q^\ell = \mathbb{E}[h_i^\ell(z)^2]$ and, for two inputs $z, z'$, correlation

$$c^\ell = \frac{\mathbb{E}[h_i^\ell(z) h_i^\ell(z')]}{\sqrt{q^\ell q^\ell}}.$$

Standard mean-field arguments (CLT plus law of large numbers) give the deterministic recursions

$$q^{\ell+1} = \sigma_w^2 \, \mathbb{E}_{g \sim \mathcal{N}(0, q^\ell)}[\phi(g)^2], \quad c^{\ell+1} = \frac{\sigma_w^2}{q^{\ell+1}} \mathbb{E}_{(g,g') \sim \mathcal{N}(0, \Sigma^\ell)}[\phi(g)\phi(g')],$$

where $\Sigma^\ell$ has entries $\Sigma_{11}^\ell = \Sigma_{22}^\ell = q^\ell$ and $\Sigma_{12}^\ell = \Sigma_{21}^\ell = c^\ell q^\ell$. For generic $(\sigma_w, \phi)$ the map $c \mapsto c^{\ell+1}$ has a unique fixed point $c_* < 1$ and $c^\ell \to c_*$ exponentially fast in $L$ for any pair of nearby but distinct inputs $z \neq z'$ (this is the ordered phase of Poole et al. (2016)).

**Jacobian structure.** The input–output Jacobian can be written as

$$J_\theta(z) = \nabla_z u_\theta(z) = v^\top \Big( \prod_{\ell=L}^1 D^\ell(z) W^\ell \Big),$$

where $D^\ell(z) = \mathrm{diag}(\phi'(h^\ell(z)))$. By independence of $v$ and the $W^\ell$ and isotropy of the weights, $\mathbb{E}\|J_\theta(z)\|_F^2$ is proportional to the squared Frobenius norm of the product

$$G^L(z) \equiv \prod_{\ell=L}^1 D^\ell(z) W^\ell$$

restricted to any fixed input direction. A standard layerwise computation (e.g. Poole et al. (2016); Balduzzi et al. (2017)) gives the recursion

$$\mathbb{E}\|G^{\ell+1}(z)\|_F^2 = \chi \, \mathbb{E}\|G^\ell(z)\|_F^2, \qquad \chi \equiv \sigma_w^2 \, \mathbb{E}_{g \sim \mathcal{N}(0, q^\ell)}[\phi'(g)^2],$$

and $\chi$ is constant along $\ell$ once $q^\ell$ has converged. Iterating yields

$$\mathbb{E}\|J_\theta(z)\|_F^2 = \Theta(\gamma^L), \qquad \gamma \equiv \chi,$$

with $\gamma = 1$ only on the edge-of-chaos manifold where $\chi = 1$. This proves (B).

**Jacobian correlations.** For two inputs $z, z'$, the same computation for the joint process gives

$$\mathbb{E}\big[\langle J_\theta(z), J_\theta(z') \rangle\big] = \chi_{12} \, \mathbb{E}\big[\langle G^{L-1}(z), G^{L-1}(z') \rangle\big],$$

with

$$\chi_{12} = \sigma_w^2 \mathbb{E}_{(g,g') \sim \mathcal{N}(0, \Sigma^*)}[\phi'(g)\phi'(g')],$$

where $\Sigma^*$ is the covariance at the fixed point $c_*$. Similarly,

$$\mathbb{E}\|J_\theta(z)\|_F^2 = \chi \, \mathbb{E}\|G^{L-1}(z)\|_F^2, \quad \mathbb{E}\|J_\theta(z')\|_F^2 = \chi \, \mathbb{E}\|G^{L-1}(z')\|_F^2.$$

Hence the cosine similarity satisfies

$$\mathbb{E}\big[\cos(J_\theta(z), J_\theta(z'))\big] \approx \Big( \frac{\chi_{12}}{\chi} \Big)^L := \rho^L,$$

up to subexponential factors from finite depth transients, where $\rho \in (0, 1)$ for $z \neq z'$ in the ordered phase since $c_* < 1$ and $\phi$ is nondegenerate. This gives exponential decorrelation and proves (A). □

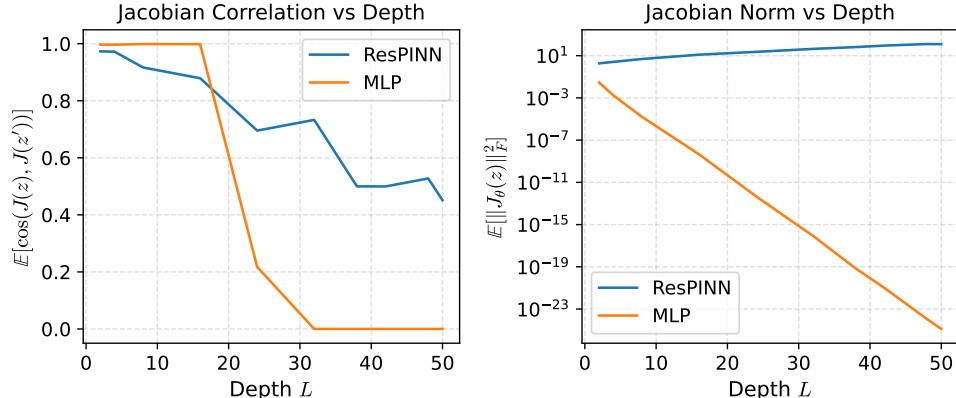

Figure 6: Left: Correlation between Jacobians. The cosine similarity of input-output Jacobians decay rapidly with depth for standard MLP PINNs. Residual PINNs maintain significantly higher gradient correlation across depth. Right: Jacobian Norms across depth(log scale). The expected Jacobian norm decreases exponentially with depth for MLP PINNs, while residual architectures maintain stable gradient magnitudes. Both plots were obtained from one common setup: fully connected and residual networks with tanh activation and width 128 were evaluated on a fixed grid of 256 points over the domain $(-1, 1) \times (0, 1)$, and all reported quantities represent averages over 20 independent random seeds.

**Empirical illustration.** The mean-field analysis predicts vanishing or exploding Jacobian norms and exponential loss of correlation across nearby inputs. Figure 5 provides an empirical counterpart: for the convection PDE, we plot the distribution of input–output Jacobian entries on a $50 \times 50$ evaluation grid. For a trained PINN, most Jacobian values concentrate near zero, indicating collapsed sensitivities, with only a few gradients reflecting the true structure of the solution. By contrast, the analytical Jacobian remains well spread, showing the expected variation across space–time. This behavior has been also observed on other PDES where PINNs exhibit failure modes.

# D   PDE Setups and Metrics

## D.1   Metrics

In our experiments, we report three metrics: the training loss (defined in Eq. (2)), the relative mean absolute error (rMAE), and the relative root mean squared error (rMSE). For a set of evaluation points $\mathcal{S}$, model prediction $u_\theta$, and ground-truth solution $u^*$, we define

$$\text{rMAE} = \frac{\sum\limits_{x \in \mathcal{S}} \left|u_\theta(x) - u^*(x)\right|}{\sum\limits_{x \in \mathcal{S}} \left|u^*(x)\right|}, \qquad \text{rMSE} = \sqrt{\frac{\sum\limits_{x \in \mathcal{S}} \left(u_\theta(x) - u^*(x)\right)^2}{\sum\limits_{x \in \mathcal{S}} \left(u^*(x)\right)^2}}. \tag{11}$$

Note that both $u_\theta(x)$ and $u^*(x)$ can take positive or negative values; consequently, rMAE and rMSE may exceed 1.

## D.2   Benchmarks

To comprehensively test our algorithm, we include four benchmarks. The first three correspond to canonical PDEs widely used in the PINN literature (see Figure 7), while the last one is the large-scale *PINNacle* benchmark Zhongkai et al. (2024).

**1D–Reaction.** This one-dimensional nonlinear ODE models chemical reactions:

$$\frac{\partial u}{\partial t} - \rho u(1 - u) = 0, \quad x \in (0, 2\pi), \ t \in (0, 1),$$

with initial and boundary conditions

$$u(x,0) = \exp\left(-\frac{(x-\pi)^2}{2(\pi/4)^2}\right), \quad u(0,t) = u(2\pi,t).$$

The analytic solution is

$$u(x,t) = \frac{h(x)e^{\rho t}}{h(x)e^{\rho t} + 1 - h(x)}, \quad h(x) = \exp\left(-\frac{(x-\pi)^2}{2(\pi/4)^2}\right),$$

with $\rho = 5$. Prior work Raissi et al. (2019); Krishnapriyan et al. (2021) identified this case as a "PINN failure mode" due to the nonlinear term, and its sharp interior boundary adds further difficulty. Following PINNsFormer Xu et al. (2025a), we sample 101 points on the initial/boundary sets and a $101 \times 101$ grid on the residual domain. Evaluation uses the same mesh.

**1D–Wave.** A standard hyperbolic PDE from acoustics and fluid dynamics:

$$\frac{\partial^2 u}{\partial t^2} - 4\frac{\partial^2 u}{\partial x^2} = 0, \quad x \in (0,1), \ t \in (0,1),$$

with initial and boundary conditions

$$u(x,0) = \sin(\pi x) + \tfrac{1}{2}\sin(\beta\pi x), \quad \frac{\partial u(x,0)}{\partial t} = 0, \quad u(0,t) = u(1,t) = 0.$$

The analytic solution is

$$u(x,t) = \sin(\pi x)\cos(2\pi t) + \tfrac{1}{2}\sin(\beta\pi x)\cos(2\beta\pi t),$$

with $\beta = 3$. Compared to Reaction and Convection, the solution is smoother, making it easier for deep models. Training/evaluation meshes are sampled as in Reaction.

**1D–Convection.** A hyperbolic PDE relevant in fluids, atmosphere, and heat transfer:

$$\frac{\partial u}{\partial t} + \beta\frac{\partial u}{\partial x} = 0, \quad x \in (0, 2\pi), \ t \in (0,1),$$

with

$$u(x,0) = \sin(x), \quad u(0,t) = u(2\pi,t).$$

The analytic solution is $u(x,t) = \sin(x - \beta t)$, where we set $\beta = 50$. Despite its simple closed form, this problem is challenging for PINNs due to the high-frequency oscillations and sharp loss landscape Krishnapriyan et al. (2021). Training/evaluation meshes follow the same setup as above.

**Heat Equation:** The heat equation is a second-order parabolic PDE that describes heat distribution in a given region over time. It is a classic example of a diffusive system, which presents challenges related to numerical stiffness.

- **Equation**: $\frac{\partial u}{\partial t} = \alpha\frac{\partial^2 u}{\partial x^2}$, with $\alpha = 0.1$.
- **Domain**: $(x,t) \in [0,1] \times [0,1]$.
- **Initial Condition**: $u(x,0) = \sin(\pi x)$.
- **Boundary Conditions**: $u(0,t) = 0$ and $u(1,t) = 0$ (Dirichlet).
- **Analytical Solution**: $u(x,t) = \sin(\pi x)e^{-\alpha\pi^2 t}$.

**PINNacle.** The fourth benchmark is *PINNacle* Zhongkai et al. (2024), built on DeepXDE Lu et al. (2021b). It comprises 20 PDE tasks covering fluid dynamics, heat conduction, nonlinear and multiscale phenomena, and high-dimensional settings. We found that several subtasks are unsolved by existing methods (e.g., Heat–2d-LT, NS–2d-LT, Wave–2d-MS, Kuramoto–Sivashinsky). These involve long-time dynamics or high-order derivatives, which present challenges beyond those noted in the original paper. To focus on training paradigms rather than backbone design, we omit these four hardest cases and evaluate on the remaining 16 tasks. Dataset details are summarized in Table 4.

Table 4: PDE benchmarks from PINNacle Zhongkai et al. (2024). We list input dimensionality, training/testing sizes, and representative simplified equations. All PDEs here are second-order. Full formalizations, coefficient meanings, and boundary/initial conditions appear in Zhongkai et al. (2024).

| PDE | Dimension | $N_{\text{train}}$ | $N_{\text{test}}$ | Key Equation |
|---|---|---|---|---|
| Burgers | 1D+Time (1d-C) | 16384 | 12288 | $\frac{\partial u}{\partial t} + u \cdot \nabla u - \nu \Delta u = 0$ |
| | 2D+Time (2d-C) | 98308 | 82690 | same form in 2D |
| Poisson | 2D (2d-C) | 12288 | 10240 | $-\Delta u = 0$ |
| | 2D (2d-CG) | 12288 | 10240 | $-\Delta u + k^2 u = f(x,y)$ |
| | 3D (3d-CG) | 49152 | 40960 | $-\mu_i \Delta u + k_i^2 u = f(x,y,z),\ i = 1,2$ |
| | 2D (2d-MS) | 12288 | 10329 | $-\nabla(a(x)\nabla u) = f(x,y)$ |
| Heat | 2D+Time (2d-VC) | 65536 | 49189 | $\frac{\partial u}{\partial t} - \nabla(a(x)\nabla u) = f(x,t)$ |
| | 2D+Time (2d-MS) | 65536 | 49189 | $\frac{\partial u}{\partial t} - \frac{1}{(500\pi)^2}u_{xx} - \frac{1}{\pi^2}u_{yy} = 0$ |
| | 2D+Time (2d-CG) | 65536 | 49152 | $\frac{\partial u}{\partial t} - \Delta u = 0$ |
| Navier–Stokes | 2D (2d-C) | 14337 | 12378 | $u \cdot \nabla u + \nabla p - \frac{1}{Re}\Delta u = 0,\ \nabla \cdot u = 0$ |
| | 2D (2d-CG) | 14055 | 12007 | same form |
| Wave | 1D+Time (1d-C) | 12288 | 10329 | $u_{tt} - 4u_{xx} = 0$ |
| | 2D+Time (2d-CG) | 49170 | 42194 | $\left[\nabla^2 - \frac{1}{c(x)}\frac{\partial^2}{\partial t^2}\right]u(x,t) = 0$ |
| Chaotic (GS) | 2D+Time | 65536 | 61780 | $\begin{cases} u_t = \varepsilon_1 \Delta u + b(1-u) - uv^2, \\ v_t = \varepsilon_2 \Delta v - dv + uv^2 \end{cases}$ |
| High-dim | 5D (P-Nd) | 49152 | 67241 | $-\Delta u = \frac{\pi^2}{4}\sum_{i=1}^n \sin\left(\frac{\pi}{2}x_i\right)$ |
| | 5D+Time (H-Nd) | 65537 | 49152 | $\frac{\partial u}{\partial t} = k\Delta u + f(x,t)$ |

## E  PINNACLE PDE BENCHMARK AND HARDER PDEs

In addition to the PINNacle results in Table 5, we evaluate ResPINN and PirateNet on several harder PDEs: Allen–Cahn, Korteweg–de Vries, Ginzburg–Landau, lid-driven cavity, and Rayleigh–Taylor instability. Both models are trained under identical conditions, and accuracy is measured using the integrated $L^2$ error over the full spatio-temporal domain. As shown in Table 6, ResPINN attains comparable or lower error across all equations, demonstrating stable performance even on more complex, nonlinear dynamics.

## F  ODE SOLVERS AND RESIDUAL FLOWS

For completeness, we recall the connection between residual updates and classical numerical ODE solvers. Consider an ODE

$$\frac{dh(t)}{dt} = f(h(t), t), \qquad h(0) = h_0.$$

### F.1  RESIDUAL FLOW SOLVERS

**Forward Euler.** The simplest explicit solver advances in steps of size $\alpha > 0$ via

$$h_{k+1} = h_k + \alpha\, f(h_k, t_k).$$

This is precisely the form of a residual block: each step applies a correction around the identity.

Table 5: Results on PINNacle. Baseline results are from Wu et al. (2024); Xu et al. (2025a). OOM means Out-of-Memory.

| | PINN | | PINNsFormer | | PINNMamba | | ResPINN | |
|---|---|---|---|---|---|---|---|---|
| Equation | rMAE | rRMSE | rMAE | rRMSE | rMAE | rRMSE | rMAE | rRMSE |
| Burgers 1d-C | 1.1e-2 | 3.3e-2 | 9.3e-3 | 1.4e-2 | 3.7e-3 | 1.1e-3 | 4.6e-3 | 1.4e-3 |
| Burgers 2d-C | 4.5e-1 | 5.2e-1 | OOM | OOM | OOM | OOM | OOM | OOM |
| Poisson 2d-C | 7.5e-1 | 6.8e-1 | 7.2e-1 | 6.6e-1 | 6.2e-1 | 5.7e-1 | 7.8e-1 | 7.1e-1 |
| Poisson 2d-CG | 5.4e-1 | 6.6e-1 | 5.4e-1 | 6.3e-1 | 1.2e-1 | 1.4e-1 | 4.4e-3 | 8.6e-3 |
| Poisson 3d-CG | 4.2e-1 | 5.0e-1 | OOM | OOM | OOM | OOM | OOM | OOM |
| Poisson 2d-MS | 7.8e-1 | 6.4e-1 | 1.3e+0 | 1.1e+0 | 7.2e-1 | 6.0e-1 | 9.0e-1 | 7.5e-1 |
| Heat 2d-VC | 1.2e+0 | 9.8e-1 | OOM | OOM | OOM | OOM | OOM | OOM |
| Heat 2d-MS | 4.7e-2 | 6.9e-2 | OOM | OOM | OOM | OOM | 6.5e-3 | 4.5e-3 |
| Heat 2d-CG | 2.7e-2 | 2.3e-2 | OOM | OOM | OOM | OOM | OOM | OOM |
| NS 2d-C | 6.1e-2 | 5.1e-2 | OOM | OOM | OOM | OOM | OOM | OOM |
| NS 2d-CG | 1.8e-1 | 1.1e-1 | 1.0e-1 | 7.0e-2 | 1.1e-2 | 7.8e-3 | 1.4e-2 | 9.8e-3 |
| Wave 1d-C | 5.5e-1 | 5.5e-1 | 5.0e-1 | 5.1e-1 | 1.0e-1 | 1.0e-1 | 3.4-2 | 3.7e-2 |
| Wave 2d-CG | 2.3e+0 | 1.6e+0 | OOM | OOM | OOM | OOM | OOM | OOM |
| Chaotic GS | 2.1e-2 | 9.4e-2 | OOM | OOM | OOM | OOM | OOM | OOM |
| High-dim PNd | 1.2e-3 | 1.1e-3 | OOM | OOM | OOM | OOM | OOM | OOM |
| High-dim HNd | 1.2e-2 | 5.3e-3 | OOM | OOM | OOM | OOM | OOM | OOM |

Table 6: Comparison of ResPINN and PirateNet on complex PDEs. Reported values are $L^2$ errors over the full spatio-temporal domain. Both models were trained and evaluated under identical settings. ResPINN attains comparable or lower error across most benchmarks, confirming that its stability extends to complex nonlinear PDEs.

| PDE | PirateNet | ResPINN |
|---|---|---|
| Allen–Cahn | $2.24 \times 10^{-5}$ | $\mathbf{2.19 \times 10^{-5}}$ |
| Korteweg–de Vries | $7.04 \times 10^{-4}$ | $\mathbf{5.05 \times 10^{-4}}$ |
| Ginzburg–Landau | $1.49 \times 10^{-2}$ | $\mathbf{4.01 \times 10^{-3}}$ |
| Lid-driven cavity (Re $= 5 \times 10^3$) | $\mathbf{3.24 \times 10^{-1}}$ | $3.69 \times 10^{-1}$ |
| Rayleigh–Taylor instab. (Ra $= 10^6$) | $\mathbf{7.32 \times 10^{-2}}$ | $9.63 \times 10^{-2}$ |

**Runge–Kutta (RK4).** Higher-order solvers reduce truncation error by evaluating $f$ at intermediate points. The classical fourth-order Runge–Kutta scheme computes

$$k_1 = f(h_k, t_k),$$
$$k_2 = f(h_k + \tfrac{\alpha}{2}k_1, t_k + \tfrac{\alpha}{2}),$$
$$k_3 = f(h_k + \tfrac{\alpha}{2}k_2, t_k + \tfrac{\alpha}{2}),$$
$$k_4 = f(h_k + \alpha k_3, t_k + \alpha),$$
$$h_{k+1} = h_k + \tfrac{\alpha}{6}(k_1 + 2k_2 + 2k_3 + k_4).$$

*ResPINNs* correspond to Euler-like discrete updates , while *O-PINNs* instantiate the continuous limit using RK4 integration with weight sharing. Implementattion details follow.

F.2    IMPLEMENTATION OF RESIDUAL FLOWS

**ResPINN (discrete residual stack).** A fixed-depth network composed of $K = 10$ residual blocks, each block containing three fully connected layers of width $64$ with a skip connection. A linear encoder maps inputs to latent space, and a single fully connected output head maps back to the PDE solution.

**O-PINN (continuous residual flow).** Uses the same residual block as the vector field $f_\theta$, but instead of stacking layers explicitly, the dynamics are integrated with RK4. This yields a continuous-depth model whose trajectory corresponds to an effectively deeper residual flow.

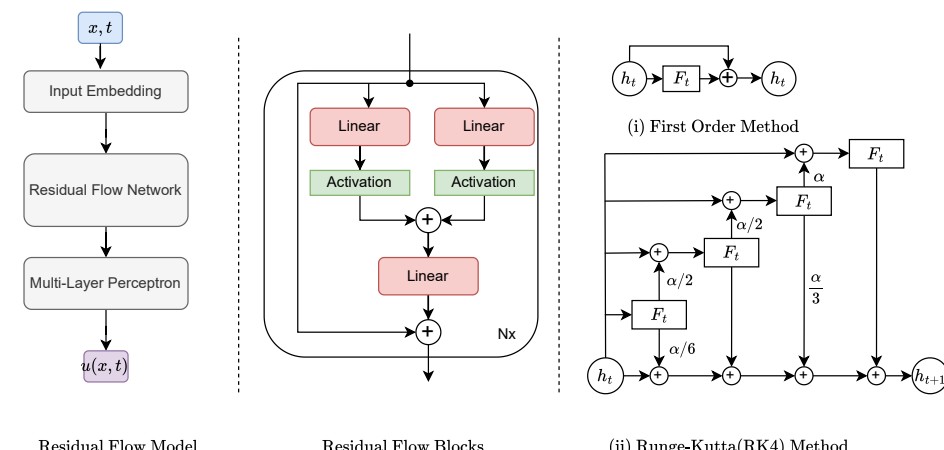

Residual Flow Model        Residual Flow Blocks        (ii) Runge-Kutta(RK4) Method

Figure 7: ResPINN overview. Inputs $(x, t)$ are encoded to a latent state $h(0)$, which is iteratively refined by a residual flow in pseudo-time $s$. The flow is realized either as a stacked residual (Euler) network or as a higher-order explicit solver RK4. The terminal state $h(S)$ is decoded to the PDE solution $u(x, t)$.

Table 7: Sensitivity to arithmetic precision. FP64 improves accuracy in PINNs at the cost of roughly doubled memory consumption. ResPINN in FP32 reaches accuracy levels comparable to FP64 PINN and exceeds it in several PDEs. Combining ResPINN with FP64 can further enhance stability and accuracy, providing additional gains where higher numerical precision is beneficial.

| PDE | rMAE | | | rMSE | | |
|---|---|---|---|---|---|---|
| | PINN64 | ResPINN32 | ResPINN64 | PINN64 | ResPINN32 | ResPINN64 |
| Wave | 0.0080 | 0.0130 | **0.0069** | 0.0081 | 0.0154 | **0.0068** |
| Reaction | 0.0271 | **0.0047** | 0.0058 | 0.0502 | **0.0075** | 0.0120 |
| Convection | 0.0059 | **0.0028** | 0.0046 | 0.0072 | **0.0046** | 0.0050 |
| Heat | **0.0003** | 0.0035 | 0.0005 | **0.0003** | 0.0048 | 0.0005 |

**Progressive Flow.** Starts with three residual blocks and adds two new blocks at each training stage while freezing earlier ones. Both encoder and decoder are linear projections, ensuring that representational capacity resides in the blocks. At each stage, the final projection layer is re-initialized and trained as a predictor of the PDE solution, providing a direct probe of iterative refinement.

An overview of ResPINN and O-PINN archirectures is shown in Figure 7.

## G   TRAINING OVERHEAD AND SENSITIVITY TO PRECISION

Several recent PINN variants introduce substantial compute and memory overhead, making it unclear whether their gains stem from architectural changes or increased resource budgets. Since our focus is on structural failure modes, we report explicit overhead comparisons and sensitivity to numerical precision. Prior work such as Xu et al. (2025b) shows that arithmetic precision can materially affect optimization, so we evaluate both FP32 and FP64. Table 8 shows that ResPINN stays lightweight: its per-epoch time is only $1.26\times$ the FP32 PINN baseline, and its peak memory use is unchanged. By contrast, transformer- and state-space–based PINNs require 6–9$\times$ more computation and 7–9$\times$ more memory, while PirateNet incurs a 3–4$\times$ overhead due to FP64 operation and added components. Table 7 shows that although FP64 stabilizes standard PINNs, ResPINN in FP32 already matches or exceeds FP64 PINN accuracy on several PDEs, with FP64 offering further gains when higher precision is helpful. These results indicate that ResPINN's improvements arise from its residual-flow structure rather than numerical precision or increased compute.

Table 8: Training Overhead and Accuracy (Wave PDE) with best computational and memory overheads highlighted in bold. ResPINN maintains accuracy improvements while remaining computationally lightweight. Its per epoch wall time and peak memory remain close to the Vanilla PINN baseline and far below sequence type models and PirateNet. The method provides these gains with only $1.26\times$ time overhead and no memory increase. Sequence models require 6 to $9\times$ more computation and 7 to $9\times$ more memory. PirateNet incurs 3 to $4\times$ overhead.

| Method | Params | Time per epoch (s) | Time Over-head | Peak Memory (MiB) | Memory Over-head | rMAE | rMSE |
|---|---|---|---|---|---|---|---|
| PINN | 527k | 1.01 | $\times 1$ | 2028 | $\times 1$ | 0.4101 | 0.4141 |
| ResPINN(ours) | **53k** | 1.27 | $\times\mathbf{1.26}$ | 2025 | $\times\mathbf{1}$ | **0.0130** | **0.0154** |
| PINNsFormer | 454k | 6.97 | $\times 6.90$ | 19170 | $\times 9.45$ | 0.3559 | 0.3622 |
| PINNMamba | 286k | 6.77 | $\times 6.70$ | 15574 | $\times 7.68$ | 0.0197 | 0.0199 |
| PirateNet | 724k | 3.12 | $\times 3.09$ | 7324 | $\times 3.61$ | 0.2544 | 0.2637 |

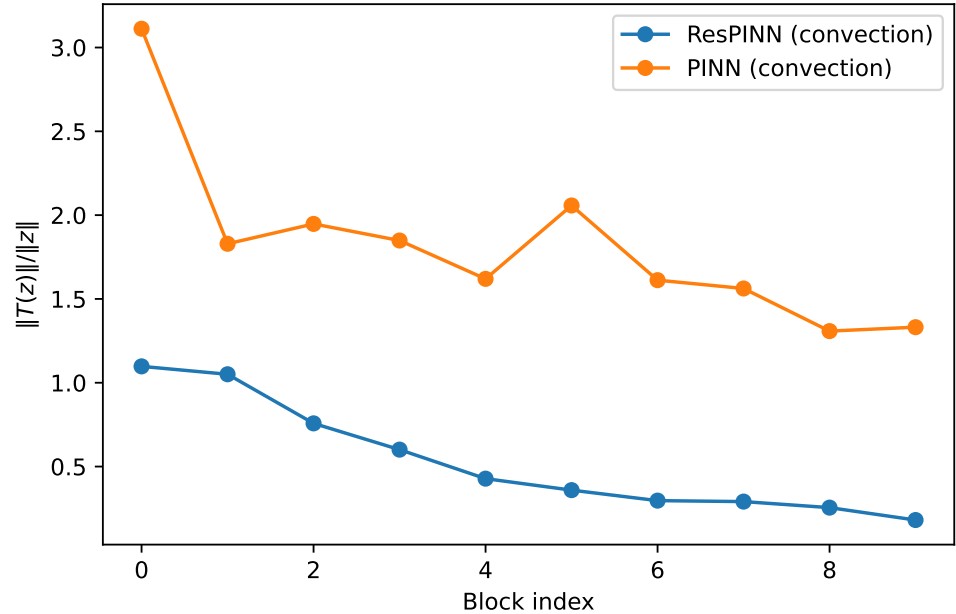

Figure 8: Relative transformation magnitude $\|T(z_k)\|/\|z_k\|$ per block for the convection problem. ResPINNs keep ratios near unity, suppressing spectral growth and stabilizing gradient flow. In contrast, PINNs amplify inputs more strongly, reflecting anisotropy and poor conditioning.

## H ADDITIONAL ALIGNMENT PLOTS

## I MORE ON ERROR AND SOLUTION MAPS

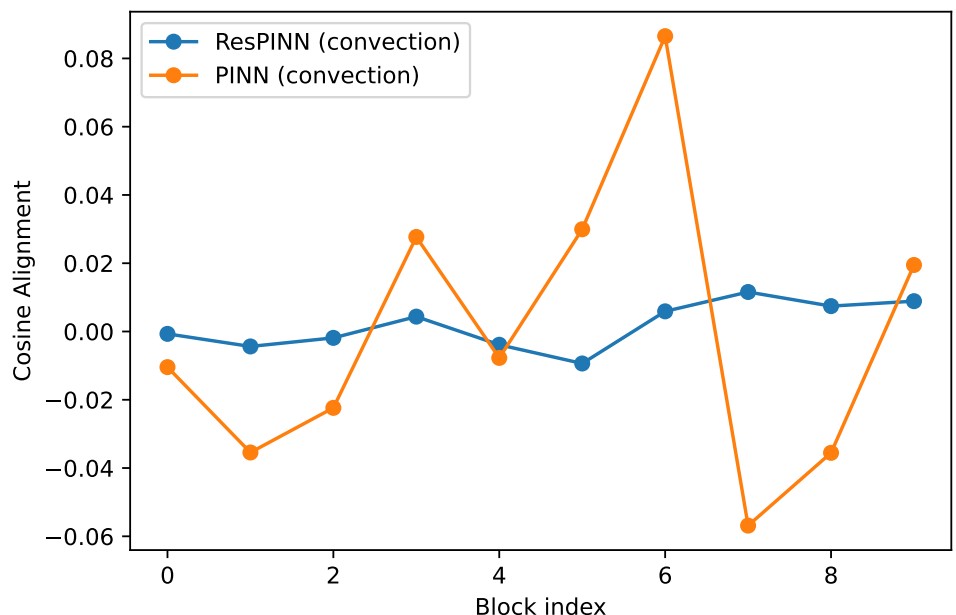

Figure 9: Cosine alignment between block updates and local loss gradients for the convection problem. ResPINNs remain close to zero, indicating residual updates act primarily as stabilizers rather than directly following descent directions. PINNs oscillate between positive and negative values, reflecting inconsistent alignment and unstable propagation.

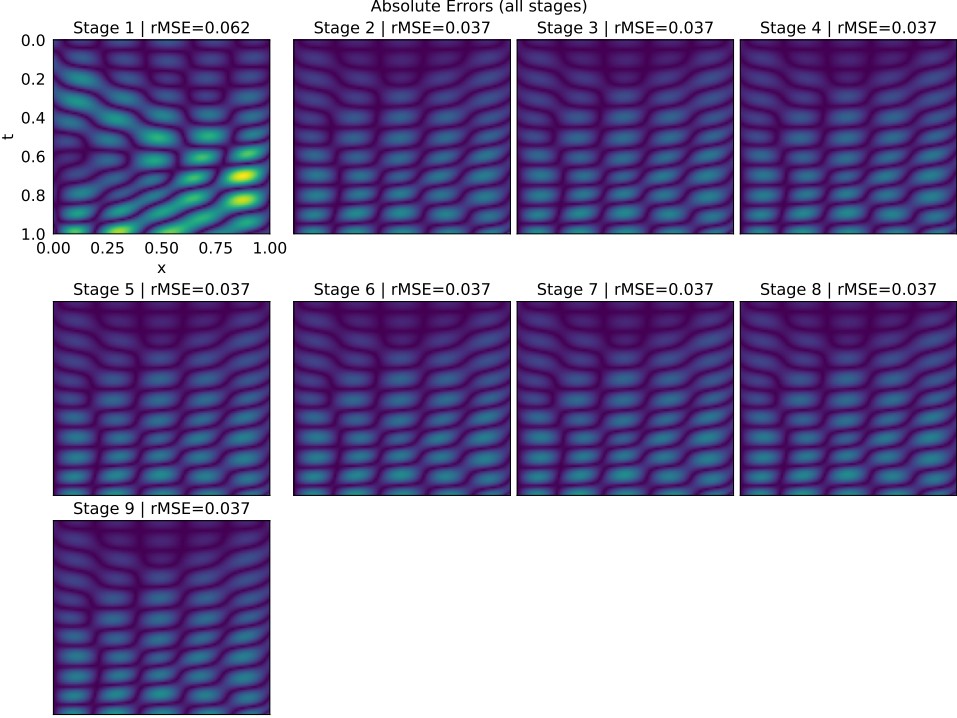

Figure 10: Absolute Errors across blocks on wave PDE.

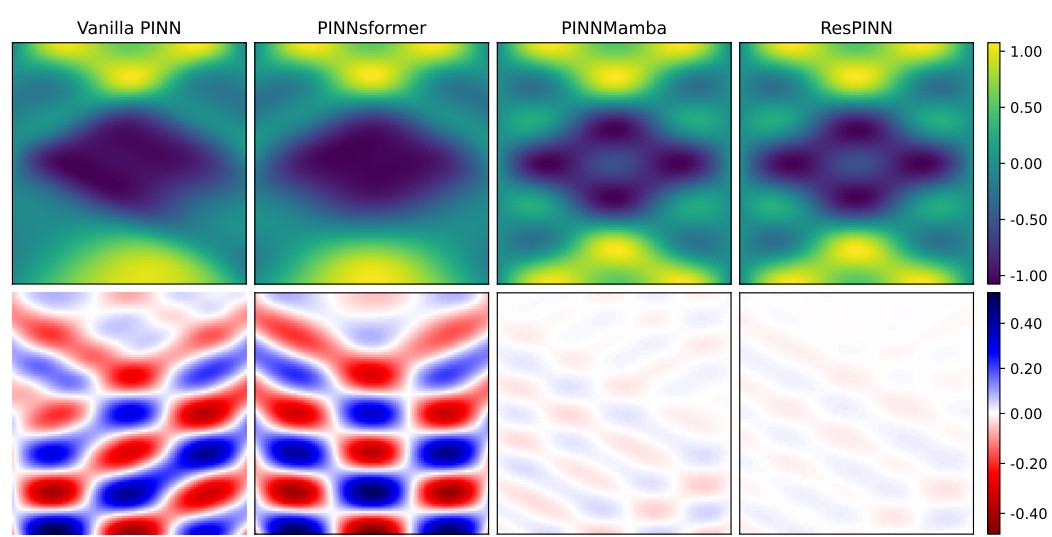

Figure 11: Qualitative comparison on 1D wave PDE. Top: predicted solutions. Bottom: pointwise errors.

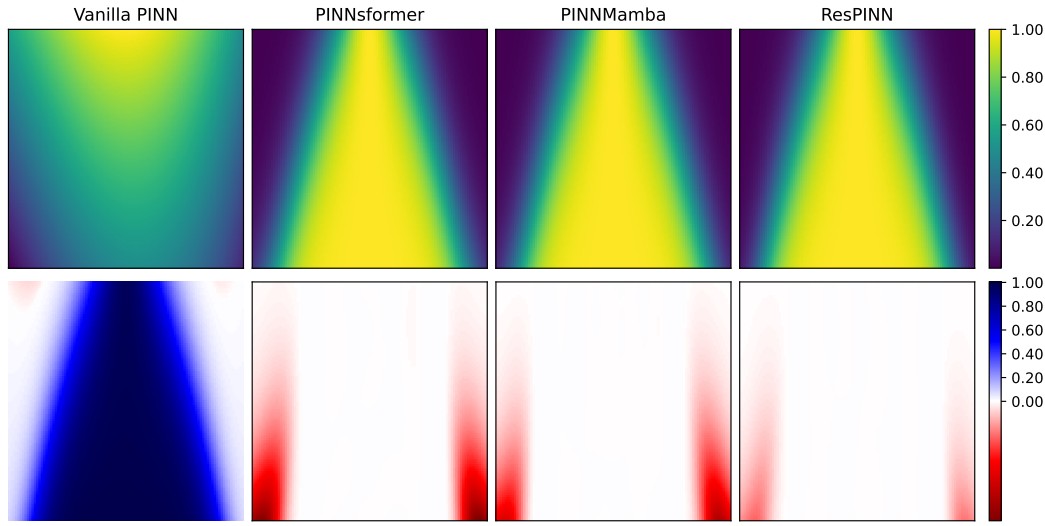

Figure 12: Qualitative comparison on 1D Reaction PDE. Top: predicted solutions. Bottom: pointwise errors.

