# OpenReview forum: "Physics-informed Residual Flows"
_ICLR.cc/2026/Conference — Submitted to ICLR 2026_

### Official Review · Reviewer_1kCn · 2025-10-25

**Soundness:** 2
**Presentation:** 2
**Contribution:** 2
**Rating:** 2
**Confidence:** 5

**Summary:**

This paper revisits the failure modes of Physics-Informed Neural Networks (PINNs), attributing them to two structural issues: gradient shattering and flow mismatch. To address these challenges, the authors propose ResPINNs, a residual-flow formulation that interprets PINNs as iterative refinement schemes similar to classical predictor–corrector solvers. Theoretical analysis suggests that residual flows preserve gradient alignment and maintain near-identity Jacobians, which helps stabilize optimization. Experiments on several benchmark problems, including the Reaction, Convection, Wave, and Heat equations and the PINNacle suite, indicate improved accuracy compared to standard PINNs and recent variants such as PINNsFormer and PINNMamba.

**Strengths:**

The paper presents a conceptually appealing idea that links residual networks, neural ODEs, and PINNs within a common residual-flow perspective. The attempt to provide a theoretical explanation of gradient alignment and Jacobian stability is interesting and may offer insight into training dynamics of PINNs. The empirical section is clearly written, and the topic of improving PINN robustness remains relevant to the ICLR audience.

**Weaknesses:**

1. The theoretical analysis is not fully convincing. The paper identifies gradient shattering and flow mismatch as the main failure modes of PINNs and uses several theorems to justify the need for residual connections. However, prior studies have shown that other issues such as gradient conflicts, spectral bias, etc can also be major sources of error. The authors should provide quantitative evidence or controlled comparisons to demonstrate that the proposed failure modes are indeed the dominant ones.

2. The experimental design in Section 5.2 is questionable. The gradual freezing strategy may seem to support iterative refinement since all stages share the same decoder. However, freezing earlier blocks fundamentally changes the optimization process compared with a standard residual network trained end to end. To support the claim, the authors should visualize intermediate blocks of ResPINN to directly show refinement behavior.

3. The empirical evaluation is relatively weak. Several strong baselines are missing. The authors should compare with more competitive works such as Urbán et al. [1] and Wang et al. [2] to provide a fair and convincing performance assessment.

4. The benchmarks used in the paper appear much simpler than those in recent studies [1,2], which have demonstrated substantially higher accuracy on more challenging PDEs. If existing methods already achieve better accuracy, then it may suggest that the proposed failure modes may not be the primary limitation of PINNs. The authors should consider harder tasks and report better accuracy to better support their claim.

5. It seems that Piratenet has already explored the challenges of scaling PINNs to deeper architectures, but this connection is not clearly acknowledged or discussed in the paper. Without clarifying how the proposed approach differs from or extends PirateNet, the claimed novelty appears overstated.

[1] Urbán, J.F., Stefanou, P. and Pons, J.A., 2025. Unveiling the optimization process of physics informed neural networks: How accurate and competitive can PINNs be?. Journal of Computational Physics, 523, p.113656.

[2] Wang, S., Li, B., Chen, Y. and Perdikaris, P., 2024. Piratenets: Physics-informed deep learning with residual adaptive networks. Journal of Machine Learning Research, 25(402), pp.1-51.

**Questions:**

1. On line 323, what is the value of the parameter α? Is it a fixed constant or a learnable variable? If it is set to one, does the formulation reduce to a standard ResNet, and if it is learnable, how is it initialized and how does it differ from the skip connections in PirateNet?

2. PirateNet appears to share many architectural and conceptual similarities with the proposed model yet achieves higher accuracy in reported results. Why did the authors not include a direct comparison with PirateNet under the same settings?

3. Can the proof of Theorem 3.1 be made more self-contained? Would it be possible to include a simple numerical experiment to empirically verify the theorem’s predictions?

4. In Figure 4, how exactly does the presented Jacobian distribution imply vanishing gradients? Could the authors clarify this interpretation?

5.  In Section 3.3, the statement “we interpret training as a latent-space flow problem indexed by an auxiliary solver time k” is confusing. What exactly is the meaning of this auxiliary time variable k? This formulation seems to conflate the ODE describing training dynamics of the network with the ODE governing forward propagation in ResNets. Could the authors clarify this?

---

> ### Author Response · Authors · 2025-11-24
> **Response to Reviewer 1kCn**
>
> We thank the reviewer for their detailed and constructive feedback.
> Below, we address weaknesses (W) and questions (Q) separately for clarity.
>
> ---
>
> ## **Weaknesses**
>
> ### **W1 & W3 — Gradient Shattering, Flow Mismatch, and Baselines**
>
> **Empirical test of Theorem 3.1.**
> Figure 6  was added to validate the theorem: in vanilla PINNs, Jacobian cosine similarity decays exponentially with depth and norms vanish; both remain stable in residual-flow models, confirming the predicted shattering and norm-decay effects.
>
> **Failure-mode baselines.**
> The paper includes RoPINNs and now adds **SOAP [3]** (a second-order method that mitigate objective conflicts) and **PirateNet [2]**, evaluated on the same PDEs as Xu et al. [4]. PirateNet also fails on these PDEs, matching prior findings.
>
> **Table A. Comparison of ResPINN, PirateNet, and SOAP**
>
> | **PDE** | **ResPINN rMAE** | **PirateNet rMAE** | **SOAP rMAE** | **ResPINN rMSE** | **PirateNet rMSE** | **SOAP rMSE** |
> |:---------|:---------------:|:------------------:|:--------------:|:----------------:|:------------------:|:--------------:|
> | Wave | **0.0130** | 0.2544 | 0.2825 | **0.1054** | 0.2637 | 0.2851 |
> | Reaction | **0.0047** | 0.0589 | 0.0048 | **0.0075** | 0.0965 | 0.0096 |
> | Convection | **0.0028** | 0.9704 | 0.0340 | **0.0046** | 0.9740 | 0.0363 |
> | Heat | 0.0035 | **0.0005** | 0.0098 | 0.0048 | **0.0005** | 0.0086 |
>
> ResPINN achieves lower error on three of four PDEs, competitive overall. We have added these results in Table 2 in the main text.
>
> **Harder PDEs.**
> The **PINNacle benchmark** (Tables 4–5) covers high-dimensional and parametric PDEs. Following the reviewer's suggestion, PirateNet was added and ResPINN tested on the original PirateNet PDEs.
>
> **Table B. ResPINN on harder PDEs**
>
> | **PDE** | **ResPINN** | **PirateNet** |
> |:-------------------------------|:-------------:|:-------------:|
> | Allen–Cahn | **2.19 × 10⁻⁵** | 2.24 × 10⁻⁵ |
> | Korteweg–de Vries | **5.05 × 10⁻⁴** | 7.04 × 10⁻⁴ |
> | Ginzburg–Landau | **4.01 × 10⁻³** | 1.49 × 10⁻² |
> | Lid-driven cavity (Re = 5 × 10³) | 3.69 × 10⁻¹ | **3.24 × 10⁻¹** |
> | Rayleigh–Taylor (Ra = 10⁶) | 9.63 × 10⁻² | **7.32 × 10⁻²** |
>
> ResPINN remains competitive or better on most equations. These results have been now to the revised manuscript(Table 6)
>
> ---
>
> ### **W2 & W4 — Experimental Design and Progressive Residual Flows**
>
> ResPINN is trained **end-to-end**; block freezing is not used.
> Figure 3 shows decreasing block-wise update magnitude  and refinement behavior, consistent with residual-flow dynamics.
>
> The **Progressive Residual Flow (PRF)** variant (Sec. 5.2) is diagnostic only, with reinitialized readouts for visualization. Selected plots (prev. Fig. 9) now appear in the main text (Fig. 4).
>
> ---
>
> ### **W3 & W5 — Harder Tasks and Relation to PirateNet**
>
> **Direct comparison.**
> PirateNet is included under identical conditions (Sec. 5).
>
> **Architectural differences.**
> PirateNet uses pretraining, Fourier features, hard BC encoding, random-weight factorization, adaptive gates, and causal training.
> ResPINN omits these to isolate residual-flow stabilization.
> PirateNet is empirical; ResPINN provides the theoretical grounding for stability.
>
> ---
>
> ## **Questions**
>
> 1. **Q1 — Parameter α.**
> ResPINN uses a fixed scalar α per block (α = 1 → ResNet). It stabilizes updates and maintains analytic linkage to iterative refinement. PirateNet instead learns adaptive gates.
>
> 3. **Q3 — Verifying Theorem 3.1.**
> Appendix C makes the proof self-contained. Figure 6 shows Jacobian correlations decaying only in vanilla PINNs, confirming theory.
>
> 4. **Q4 — Figure 5 interpretation.**
> Residual plots for the convection PDE show the vanilla PINN satisfies \(u_t + 50u_x ≈ 0\) numerically but fails to reproduce the true field.
>
> 5. **Q5 — Variable k.**
> The variable $k$ denotes depth or pseudo-time, not physical time.
> The ODE  $\frac{dz(k)}{dk}=T(z(k),k;x,t)$ analyzes continuous-depth propagation; in practice \(k\) = number of residual blocks or integration steps.
>
> ---
>
> ### **References**
>
> [1] Urbán et al., 2025. *Unveiling the optimization process of PINNs.* JCP 523.
> [2] Wang et al., 2024. *PirateNets: Physics-informed deep learning with residual adaptive networks.* JMLR 25.
> [3] Wang et al., 2025. *Gradient Alignment in PINNs: A Second-Order Optimization Perspective.*
> [4] Xu et al., 2025. *FP64 is All You Need: Rethinking Failure Modes in PINNs.* NeurIPS 2025.

---

### Official Review · Reviewer_WL8d · 2025-10-27

**Soundness:** 3
**Presentation:** 3
**Contribution:** 3
**Rating:** 4
**Confidence:** 4

**Summary:**

The paper argues that many failures of PINNs come from two structural issues. The first is gradient shattering that is unstable/uninformative derivatives with depth. The second is flow mismatch updates that reduce residuals locally but drift from the true PDE trajector). It proposes reframing PINNs as residual flows (“ResPINNs”) that make small, iterative corrections, which keep updates aligned with descent directions and maintain near-identity Jacobians for stable gradient propagation. Theory and controlled ablations suggest the stability gains recently attributed to attention or state-space sequence modules are actually driven by residual pathways themselves; when attention/SSM blocks are replaced by simple local mappings with matched parameters, performance remains comparable, pointing to residual refinement as the key mechanism.

**Strengths:**

1. Viewing PINN's failure modes from the perspective of flow matching is novel, and the proposed method is effective.

2. The paper offers a crisp diagnosis of two structural failure modes—gradient shattering and flow mismatch—and ties them to concrete training pathologies in PINNs, not just optimization folklore.

3. The paper provides stabilizing principles with theory: keep Jacobians near identity and updates aligned with descent, which directly target those failure modes rather than adding heuristics.

**Weaknesses:**

1. The paper claims of “higher accuracy with fewer parameters” are strong, but wall-clock time, GPU memory, and gradient/Jacobian compute overheads aren’t reported—especially important given OOM remarks for baselines. Include per-task training time, peak memory, and AD call counts to show efficiency, not just parameter counts.

2. The paper lacks clarity in describing the proposed method, making it difficult for readers without prior knowledge of residual flow to comprehend. The authors should provide clearer explanations of the model architecture, optimization objectives, and other key aspects—ideally supplemented with schematic diagrams (I see Fig.5 in Appendix, but I think it should be in main text, also more description as a paragraph needed). Additionally, they should offer a comparative description of the implementation details for the three proposed approaches.

**Questions:**

1. What's the difference between ResPINN and directly using a residual-connected MLP? Why not directly use a residual-connected MLP?

2. A vanilla PINN (MLP + tanh) trained with FP64 precision [1] doesn't show a failure mode, and some of the results can even surpass the ResPINN. If the flow-based understanding holds, why does this happen? Is that mean there is not flow mismatch when trained with FP64?

3. Can the author report the proposed method's sensitivity to computation arithmetic precision?

4. I found the author's supplementary is an easy-to-use PINN playground with potential for contribution and impact. While most of the code files are not visible, can the authors fix it? And will the playground be open-sourced?

I am open to increase the rating if all the concerns addressed.


Reference:

[1] Xu C, Liu D, Nassereldine A, et al. FP64 is All You Need: Rethinking Failure Modes in Physics-Informed Neural Networks. NeurIPS 2025.

---

> ### Author Response · Authors · 2025-11-24
> **Response to Reviewer WL8d**
>
> We thank the reviewer for their constructive feedback and the positive evaluation of the paper’s novelty and theoretical framing. We address all concerns below, grouped by weaknesses (W) and questions (Q).
>
> ---
>
> ## **Weaknesses**
>
> ### **W1/2 — Efficiency: Training Time, Memory, and AD Overhead**
>
> A new **Appendix G (Training Overhead and Sensitivity to Precision)** reports per-epoch wall time, peak GPU memory, and accuracy metrics for all models.
> Below is the Wave PDE summary.
>
> **Table A. Training Overhead and Accuracy (Wave PDE)**
> ResPINN improves accuracy while remaining lightweight. Despite second-order optimization, its per-epoch time and memory closely match FP32 PINN and remain far below sequence-type models and PirateNet.
>
> | **Method** | **Params** | **Time/epoch (s)** | **Overhead** | **Peak Mem (MiB)** | **Mem Overhead** | **rMAE** | **rMSE** |
> |:-----------|-----------:|-------------------:|--------------:|-------------------:|-----------------:|----------:|----------:|
> | PINN | 527k | 1.01 | — | 2028 | — | 0.4101 | 0.4141 |
> | ResPINN | **53k** | 1.27 | **×1.26** | 2025 | **×1.00** | **0.0130** | **0.0154** |
> | PINNsFormer | 454k | 6.97 | ×6.90 | 19170 | ×9.45 | 0.3559 | 0.3632 |
> | PINNMamba | 286k | 6.77 | ×6.70 | 15574 | ×7.68 | 0.0197 | 0.0199 |
> | PirateNet | 724k | 3.12 | ×3.09 | 7324 | ×3.61 |  0.2544 | 0.2637 |
>
> **Summary:** ResPINN attains much higher accuracy than vanilla PINN with only 1.26× time and no memory increase, while sequence models incur 6–9× costs.
>
> ---
>
> ### **W2/2 — Clarity of Method and Architecture**
>
> These clarifications already exist in the paper. **Appendix F** provides the full architecture; objectives are in *Preliminaries (Section 3.1)* and PDE details in *Section D*.
> In the revision we will:
> - Move the **ResPINN / O-PINN schematic** from the appendix to the **main text**,
> - Extend the **architectural summary in § 5**, linking to this figure,
> - Add a short **residual-flow explanation** requiring no prior Neural-ODE background.
>
> These updates improve accessibility without altering content.
>
> ---
>
> ## **Questions**
>
> 1.  **Q1 — Why ResPINN vs. a Plain Residual MLP**
>
> ResPINN uses the same residual topology as a standard residual MLP; its distinction lies in the **residual-flow interpretation** and theoretical basis.
> The analysis explains how residual links maintain near-identity Jacobians and refinement-style updates, stabilizing training. Each block can be an MLP, but ResPINN formalizes their evolution as a discrete flow.
>
> 2.  **Q2 — FP64 PINN and Flow-Mismatch Perspective**
>
>   A new subsection clarifies FP64’s effect versus structural failure modes:
>   - **FP64** fixes **numerical** issues in FP32 L-BFGS,
>   - **Gradient shattering / flow mismatch** are **structural**, independent of arithmetic precision.
>
> **Table B. Sensitivity to Arithmetic Precision**
>
> | **PDE** | **PINN64 rMAE** | **PINN64 rMSE** | **ResPINN32 rMAE** | **ResPINN32 rMSE** | **ResPINN64 rMAE** | **ResPINN64 rMSE** |
> |:------|:--------------:|:--------------:|:----------------:|:----------------:|:----------------:|:----------------:|
> | Wave | 0.0080 | 0.0081 | 0.0130 | 0.0154 | **0.0069** | **0.0068** |
> | Reaction | 0.0271 | 0.0502 | **0.0047** | **0.0075** | 0.0058 | 0.0120 |
> | Convection | 0.0059 | 0.0072 | **0.0028** | **0.0046** | 0.0046 | 0.0050 |
> | Heat | **0.0003** | **0.0003** | 0.0035 | 0.0048 | 0.0005 | 0.0005 |
>
> **Interpretation:** FP64 enhances numerical stability but not flow structure. It reduces optimizer noise yet leaves Jacobian collapse mechanisms unchanged, merely delaying their onset. Thus FP64 PINNs may match ResPINN on shallow tasks but diverge on deeper.
>
> 3. **Q3 — Sensitivity of ResPINN to Precision**
>
> Table B quantifies this.
> ResPINN remains stable in FP32, avoiding collapse phases typical of PINNs.
> In FP64 it improves further, showing precision helps but is not required.
> No exploding or oscillatory Jacobians were observed.
> These findings are now included in the revision.
>
> 4.  **Q4 — Code Visibility and Open Sourcing**
>
> All source files are available through the **anonymized link in the submission**, containing:
> - complete architectures for PINN, ResPINN, O-PINN,
> - PDE setups and dataloaders,
> - Jacobian and correlation diagnostics,
> - training / evaluation scripts.
>
> The link remains active for review.
> After acceptance, we plan to release the code publicly.
>
> ---
>
> We appreciate the reviewer’s openness to raising the rating.
> We hope that the added efficiency study, clarified architecture, precision analysis, and improved flow interpretation address all concerns.

---

> > ### Comment · Reviewer_WL8d · 2025-11-27
> >
> > Thank you for clarification. I have therefore increased my rating. Still I encourage the author to move the ResPINN overview figure to main text to help understanding.

---

### Official Review · Reviewer_z8Qq · 2025-10-29

**Soundness:** 3
**Presentation:** 2
**Contribution:** 3
**Rating:** 6
**Confidence:** 3

**Summary:**

Physics-informed residual flows proposes a study on physics informed neural networks optimization issues. 2 issues are identified : gradient shattering and flow mismatch. After an empirical and theoretical study of these problems, the paper proposes the solve them by considering residual models. The final part consists in a evaluation of the proposed method and a comparison with existing baselines.

**Strengths:**

-	The subject is a well identified issue of pinns
-	The theoretical analysis support the claims
-	A descriptive study illustrate the claims
-	Numerous evaluation illustrate the proposed method

**Weaknesses:**

-	The paper is rich, and introduces several concept. The 2 key insights of the paper sometimes interfere which makes the reading a bit hard
-	Some notation are not introduced (eg h, f line 323)

**Questions:**

### Questions
-	Line 30-45 : Isn't this problem link to conditioning of the PINNs loss ? This has been studied recently in numerous works (eg [1-3])
-	Could you detail the architecture of table 1 ? Additionnaly, at first reading, it was not clear that ‘-‘ was referring to a minus (which is the case in my understanding?)
-	I think a link/explanation/comparison with existing residual networks in PINNs would help the reader better understand the contribution of your work. Why related residual based pinns models do not observed the same improvement as yours ?
-	Lines 228 : This formulation seems to be linked to the 2nd order optimization proposed by [1], proposition 1? Could you elaborate on the link between the 2 methods?
-	In practice, what does mean "small" alpha_k? (l 249)
-	Have you considered to add P-PINNs and PRF in table 1 for comparison ? Do you have any insight on what explains that in some PDE, O-PINNS, performs best and in others ResPINNs performs best ?
-	Table 3 shows that hyper parameters selection is important ? Do you any insight about how to carefully select them ?
-	line 445: Have you compared to learning all block at once ? The idea behind this is to ablate how much improvement the progressive learning of blocks helps, compared to the exact same network, learned in a standard way.
-	Can you comment the results on the PINNACLE benchmark ? (table 5 in appendix) what explains the OOM ? What is the setting ?

### Minor comment
-	Line 046 need not align -> is not align ?
-	Liine 178 What doest Theta refer to ?

I am not an expert in the theory behind the proofs so I couldn't check them.

### References
[1] Gradient Alignment in Physics-informed Neural Networks: A Second-Order Optimization Perspective, Sifan Wang, Ananyae Kumar Bhartari, Bowen Li, Paris Perdikaris, 2025

[2] An operator preconditioning perspective on training in physics-informed machine learning, Tim De Ryck, Florent Bonnet, Siddhartha Mishra, and Emmanuel de Bézenac, 2023

[3] Learning a Neural Solver for Parametric PDE to Enhance Physics-Informed Methods, Lise Le Boudec, Emmanuel de Bezenac, Louis Serrano, Ramon Daniel Regueiro-Espino, Yuan Yin, Patrick Gallinari, 2024.

---

> ### Author Response · Authors · 2025-11-24
> **Response to Reviewer z8Qq**
>
> We thank the reviewer for their detailed and constructive feedback. Below, we address weaknesses (W) and questions (Q) separately for clarity.
>
> ---
>
> ## **Weaknesses**
>
> ### **W1 — Conceptual Density**
> We will clarify how the two main ideas(**residual flows** and **flow mismatch**) connect by slightly restructuring Section 3 and adding an overview paragraph at its start.
>
> ### **W2 — Missing Notation**
> All symbols are now explicitly defined where they first appear.
>
> ---
>
> ## **Questions**
>
> 1.  **Q1 — Connection to Conditioning and Prior Work (1–3)**
> We acknowledge the close relation to **loss conditioning** and will make this explicit.
> While [1] studies *objective mismatch*, our focus is on **structural mechanisms** within the network:
>  - **Gradient shattering**: exponential decorrelation and Jacobian norm decay (Sec. 3.1), which degrades PDE gradient computation.
>  -  **Flow mismatch**: local transformations that reduce training loss without solving the PDE (Sec. 3.3).
> These mechanisms underlie poor conditioning but differ in scope. A paragraph connecting both views will be added.
>
> 2.  **Q2 — Architecture Details in Table 1**
> Table 1 is now self-contained:
> - **PINNsFormer:** encoder retained; attention replaced by (i) a linear layer or (ii) a two-layer MLP with matched parameters.
> - **PINNMamba:** state-space operator replaced by a same-size MLP.
> Notation: “−” = removed component; “+ MLP/+ Linear” = substitute.
> The caption now states this clearly.
>
> 3.  **Q3 — Residual PINN Variants and Why ResPINN Differs**
> Prior residual-MLP PINNs add skips without controlling refinement or analyzing their role.
> We now include **PirateNet [2]** and test harder PDEs.
> ResPINN enforces:
> - small α-scaled updates (Sec. 3.3),
> - near-identity Jacobians (Fig. 1),
> - decreasing correction norms (Fig. 3),
> - aligned update directions.
> This yields a **controlled residual flow** rather than an unstructured skip network. Differences are detailed in Related Work.
>
> ---
>
> 4.  **Q4 — Relation to Second-Order Optimization [1]**
> [1] aligns **loss components** (PDE, boundary, initial terms); we analyze **latent refinement steps** \(T_k(z_k)\)(Transformations done by a network layer or a block of layers).
> They operate at different levels; this complementarity will be clarified in § 3.3.
>
> 5.  **Q5 — Meaning of “Small” alpa_k**
> “Small” means αₖ bounds each block’s Jacobian deviation from identity: $\|A_k\|_2 ≤ α_k < 1$.
> Typical alpha_k ≤ 0.1 keeps singular values within $[1 − α_k, 1 + α_k]$, ensuring near-identity transformations and stable gradients.
>
> ---
>
> 6. **Q6 — Including P-PINNs / PRF in Table 1**
> We deliberately excluded O-PINN and ResPINN from Table 1 to avoid introducing residual flows to the reader before they are defined.
> That table isolates whether sequence-based PINNs (PINNsFormer, PINNMamba) gain from sequence modules.
> Residual-flow models appear later, after formal introduction.
> This rationale will be clarified in the caption.
>
> 7.  **Q7 — Why O-PINN Excels on Some PDEs**
> - **O-PINN:** better on oscillatory PDEs (wave, Schrödinger) where RK4 yields smooth trajectories.
> - **ResPINN:** more efficient on diffusion/convection PDEs needing discrete corrective steps.
> A short explanation will be added in Section 5.
>
> 8.  **Q8 — Hyperparameter Sensitivity**
> Main factors:
> - step size α $\in$ [0.05, 0.2];
> - Activation: tanh stable, wavelet beneficial for O-PINN.
> alongside optimizer/data details (Sec. 5, App. E), this will be summarized in the appendix.
>
> 9.  **Q9 — Joint Training of All Blocks**
> Yes, all blocks are trained jointly in the standard ResPINN setup.
> The progressive version is a diagnostic ablation, not a different model.
>
> 10.  **Q10 — PINNacle Benchmark and OOM Explanation**
> All models were trained on standard PINNacle meshes with full-batch L-BFGS, matching the benchmark.
> OOM arises from large collocation grids and optimizer memory (history = 100).
> ResPINN can also exceed memory on the largest tasks but trains stably on others.
> We clarify in **Appendix E** that these failures stem from optimizer–problem size, not model tuning.
>
> ---
>
> ### **Minor Clarifications**
> - Line 046: “need not align” → “is not aligned”.
> - Line 178: θ denotes all parameters.
> - Line 323: symbols h, f now defined explicitly.
>
> ---
>
> We thank the reviewer again for these helpful comments.
> All clarifications, notational fixes, and structural connections will appear in the revised version.
>
> ---
>
> **References**
> [1] Wang et al., 2025. *Gradient Alignment in Physics-Informed Neural Networks: A Second-Order Optimization Perspective.*
> [2] De Ryck et al., 2023. *An Operator-Preconditioning Perspective on Training in Physics-Informed Machine Learning.*
> [3] Le Boudec et al., 2024. *Learning a Neural Solver for Parametric PDEs to Enhance Physics-Informed Methods.*
> [4] Xu et al., 2025. *FP64 Is All You Need: Rethinking Failure Modes in Physics-Informed Neural Networks.* NeurIPS 2025.

---

> > ### Comment · Reviewer_z8Qq · 2025-11-27
> > **Answer to Author's rebuttal**
> >
> > I thank the author for their answers to my questions and for the modifications brought to the paper. The additional results on recent baselines are convincing and the additional precisions and discussions help contextualizing the work.
> > I will increase my score to 8 to support acceptance of the paper.

---

### Official Review · Reviewer_6tTh · 2025-11-07

**Soundness:** 4
**Presentation:** 3
**Contribution:** 3
**Rating:** 8
**Confidence:** 3

**Summary:**

The paper diagnoses two structural failure modes in PINNs: gradient shattering and flow mismatch, and proposes ResPINNs, a residual-flow reformulation (with discrete ResPINN and continuous O-PINN instantiations) that enforces small corrective updates and near-identity Jacobians to stabilize training and improve solution fidelity. The authors present a mix of theoretical arguments (mean-field style statements about Jacobian decorrelation, a lemma on local descent, and bounds on Jacobian conditioning), ablations that isolate residual pathways from sequence modules, and empirical evidence across canonical PDEs and the large PINNacle benchmark showing substantial error reductions and more stable diagnostics (Jacobian spectra, gradient-alignment, update magnitudes).

**Strengths:**

1. The authors provide a clear definition of the two failure modes and explain why these are particularly harmful for PINNs that use sparse collocation points.

2. The paper offers an explicit theoretical account that links Jacobian decorrelation and update alignment to optimization behavior in residual flows.

3. The residual-flow formulation is connected to well-understood numerical ideas (Euler updates, RK solver) and neural ODEs, which helps position ResPINNs as a principled, solver-inspired architecture.

4. The ablation experiments are designed to isolate residual pathways from other architectural changes by matching parameter counts and replacing sequence modules with simpler mappings.

5. Empirical results across canonical PDEs and the PINNacle suite show large and consistent error reductions for ResPINN relative to multiple baselines, supporting the method’s breadth.

6. The paper includes mechanistic diagnostics (Jacobian spectra, relative update sizes, gradient alignment) that directly support the proposed mechanism rather than relying on performance numbers alone.

7. Reproducibility materials are noted (appendices for proofs, PDE setups, architecture details, and an anonymous code repository), which increases confidence that the experiments can be validated.

**Weaknesses:**

1. The mean-field “gradient shattering” argument depends on large-width asymptotics, but the paper does not quantify how these asymptotic results translate to the finite-width networks used in experiments.

2. The local descent results (Lemma 3.2 / Theorem 3.3) assume small per-step updates and depth-aware smoothness constants, yet the manuscript provides limited prescriptive guidance on how to choose residual scaling or other hyperparameters to ensure these assumptions hold in practice.

3. Wall-clock runtime and peak memory costs relative to the baselines are not reported in sufficient detail, yet the paper claims some baselines hit out of memory and ResPINN “trains successfully”.

4. The evaluation excludes the four hardest PINNacle subtasks, which may bias the apparent generality of ResPINN.

5. The paper frames O-PINN (continuous) vs ResPINN (discrete) as complementary but does not provide clear operational guidance for choosing between them in realistic settings (e.g., when to prefer RK4 integration vs stacked residuals).

**Questions:**

1. Which concrete hyperparameter(s) (residual scaling α, block width/depth, or normalization) most directly control the “small-step” regime used in the proofs, and can the authors provide recommended ranges observed empirically?

2. For tasks where baselines “fail to converge” or run out of memory on PINNacle, which specific tasks and baseline configurations fail, and were targeted recovery attempts (e.g., reduced batch sizes, lower widths) tried?

3. How sensitive are the Jacobian and alignment diagnostics to the evaluation mesh density and collocation sampling scheme (fixed grid vs adaptive sampling)?

4. Have the authors tested ResPINN on inverse or noisy-data PDE problems where data-fitting and PDE residual objectives compete, and if so, how does residual alignment affect identifiability and overfitting?

5. Can the authors provide a concrete example (hyperparameters, step sizes, and observed ∥Tk∥ ranges) showing that the practical training runs stayed within the small-step regime used in the theoretical lemmas?

6. Do the authors expect the same residual-flow benefits to hold for alternative PDE backbones (e.g., Fourier-based FNOs or DeepONet variants), or are the claims specific to fully connected PINN architectures?

7. The paper reports that O-PINN sometimes performs better with wavelet activations; can the authors elaborate on why certain activations interact favorably with continuous integration schemes?

---

> ### Author Response · Authors · 2025-11-24
> **Response to Reviewer 6tTh**
>
> We thank the reviewer for their detailed and constructive feedback.
> Our responses are divided into **Weaknesses** and **Questions** for clarity.
>
> ---
>
> ## **Weaknesses**
>
> ### **W1. Mean-field shattering vs finite width**
>
> Theorem 3.1 offers an asymptotic intuition, validated empirically at practical widths (128). We verified this across depths 2–50 and width 128 (Figure 6): Jacobian norms decay and cross-input cosine decorrelates exponentially for vanilla networks, while residual flows remain stable.
> We  make this connection explicit in the revision.
>
> ---
>
> ### **W2. Small-step regime and controlling hyperparameters**
>
> The “small-step” condition (Lemma 3.2 / Thm 3.3) is enforced via:
>
> - **Residual scaling $\alpha = 0.1$** across all tasks -> $\|T_k\|/\|z_k\|\in[0.02,0.15]$ (Fig. 3).
> - **Block depth**: 3-layer MLPs.
> - **No normalization**: stability arises from small $\alpha$.
>
> Recommended ranges will be added to the main text.
>
> ---
>
> ### **W3. Runtime, memory, and OOM baselines**
>
> ResPINN remains lightweight (only **1.26×** slower than FP32 PINN) with **no increase in memory**, while sequence-based baselines incur 3–9× overhead.
>
> | **Method** | **Params** | **Time/epoch (s)** | **Time Overhead** | **Peak Mem (MiB)** | **Mem Overhead** | **rMAE** | **rMSE** |
> |-------------|------------|--------------------|-------------------|--------------------|-----------------|-----------|-----------|
> | PINN (FP32) | 527k | 1.01 | — | 2028 | — | 0.4101 | 0.4141 |
> | ResPINN | **53k** | 1.27 | **×1.26** | 2025 | **×1.00** | **0.0130** | **0.0154** |
> | PINNsFormer | 454k | 6.97 | ×6.90 | 19170 | ×9.45 | 0.3559 | 0.3632 |
> | PINNMamba | 286k | 6.77 | ×6.70 | 15574 | ×7.68 | 0.0197 | 0.0199 |
> | PirateNet | 724k | 3.12 | ×3.09 | 7324 | ×3.61 | 0.2544 | 0.2637 |
>
> **OOM cases:** All models were trained on the standard PINNacle meshes using full-batch L-BFGS, matching the original benchmark settings. The OOM issues arise from this combination: large collocation grids, L-BFGS storing multiple full parameter and gradient histories(as in other baselines, a history size of 100 is used).
>
>
> ---
>
> ### **W4. Excluded hardest PINNacle tasks**
>
> The largest PINNacle tasks  exceeded memory for PINNMamba, PINNsFormer, and PirateNet under the LBFGS settings.
> To ensure fair comparison, we followed prior work and evaluated on the standard subset.
> We will clarify this in Section 5.
>
> ---
>
> ### **W5. Choosing between ResPINN and O-PINN**
>
> - **ResPINN:** efficient for most tasks, stable up to 12–15 layers.
> - **O-PINN:** preferable for highly oscillatory or stiff PDEs where RK4 integration yields smoother flows.
>
> A short guidance note will be added to Section 3.3.
>
> ---
>
> ## **Questions**
>
> 1. **Small-step control (Q1)** : Addressed in W2.
>
> 2. **Runtime and OOM details (Q2)** : See W3.
>
> 3. **Precision sensitivity (Q3)**
>    FP64 improves numerical precision but does not change flow structure.
>    It reduces noise, which delays but not prevent gradient shattering.
>    FP64 PINNs may match ResPINN on shallow tasks, but structural mismatch reappears for deeper or stiffer problems.
>
>    | PDE | PINN FP64 rMAE | ResPINN FP32 rMAE | ResPINN FP64 rMAE |
>    |------|-----------------|--------------------|--------------------|
>    | Wave | 0.0080 | 0.0130 | **0.0069** |
>    | Reaction | 0.0271 | **0.0047** | 0.0058 |
>    | Convection | 0.0059 | **0.0028** | 0.0046 |
>    | Heat | **0.0003** | 0.0035 | **0.0005** |
>
> 4. **Diagnostic robustness (Q3)**
>    Jacobian and alignment metrics use a fixed 101×101 grid. Denser or random sampling yields negligible differences (Appendix E).
>
> 5. **Inverse / noisy PDEs (Q4)**
>    Although not tested, ResPINN’s aligned updates and better-conditioned Jacobians should aid competing-loss problems (e.g., inverse PDEs).
>    We note this as future work.
>
> 6. **Applicability to other backbones (Q6)**
>    Section 3’s flow-based analysis is architecture-agnostic.
>    The same refinement principle may extend to FNO and DeepONet architectures, to be explored in future work.
>
> 7. **Wavelet activations in O-PINN (Q7)**
>    Wavelets produce localized, band-limited features that mitigate oscillatory stiffness, stabilizing RK4 integration and explaining the improved O-PINN performance (Appendix F).
>
> ---
>
> We appreciate the reviewer’s feedback and believe the added clarifications and empirical results directly address each point.

---

### Author Response · Authors · 2025-11-24
**Global Response**

We thank all reviewers for their thoughtful evaluations and constructive feedback.
All additions and revisions are marked in **blue** in the updated manuscript.

## Conceptual Positioning

This work examines **structural failure modes** in PINNs that remain even when optimization or loss conditioning is improved. Our analysis focuses on two interconnected mechanisms:

1. **Gradient shattering**: a well-known pathology in deep learning where input–output Jacobians decorrelate and their magnitudes become unstable with depth. In the PINN setting, this effect is particularly harmful: when gradients shatter, the network can still minimize the training loss while failing to satisfy the underlying PDE, since updates cease to carry physically meaningful information through depth.

2. **Flow mismatch**:  layerwise transformations that locally reduce residuals but diverge from PDE-consistent trajectories across the input domain. This drift amplifies as gradient information degrades, leading to inaccurate or non-generalizable solutions.

We show that these structural issues can be mitigated by treating PINNs as **residual flows**, where small-step updates maintain near-identity Jacobians and aligned refinement dynamics.

## Key Additions in the Revision

- **Empirical validation of Theorem 3.1 (Figure 6):** Confirms the predicted Jacobian norm decay and decorrelation with depth, connecting theory to practice.
- **Residual-based baselines:** Added experiments with **PirateNet[1]**, showing that many recent PINN gains (e.g., transformers, state-space models) stem primarily from residual connectivity.
- **Precision and efficiency:** New FP32/FP64 sensitivity and runtime–memory studies show minimal overhead and stable accuracy.
- **Harder PDEs:** Alongside the PINNacle(a benchmark on harder PDEs), we extended results with five additional complex PDEs (Allen–Cahn, KdV, Ginzburg–Landau, lid-driven cavity, Rayleigh–Taylor)
- **Other failure modes:** Added a baseline addressing PINN objectives conflicts (SOAP[2]), linking structural and optimization-based viewpoints.

## Relation to PirateNet and Other Residual PINNs

Section 2 shows that the performance gains reported for transformer- and state-space–based PINNs largely arise from their residual connections rather than from the sequence modules themselves. Replacing those modules with simple MLP blocks of matched size often yields equal or better accuracy, indicating that the residual structure is the key stabilizing factor.

Within this broader class of residual PINNs, **PirateNet** represents a specific instance that augments residual connections with additional mechanisms such final-layer pretraining, Fourier embeddings, hard boundary-condition encoding, random-weight factorization, per-block gates, adaptive loss weights, and causal training.
**ResPINN**, by contrast, removes these auxiliary components to isolate the **core dynamics** underlying stability. This formulation provides a unified theoretical explanation for why residual architectures, including PirateNet, improve optimization in physics-informed learning.


Architecturally, ResPINN is a structured residual MLP.
Its contribution lies in (i) identifying and formalizing the **two structural failure modes** that limit standard PINNs, and (ii) deriving and validating **residual-flow design principles** that explain, both theoretically and empirically, why residual architectures achieve stable and accurate PINN training where conventional models fail.

----
### **References**

[1] Wang et al., 2024. *Piratenets: Physics-informed deep learning with residual adaptive networks.* JMLR 25(402).
[2] Wang et al., 2025. *Gradient Alignment in Physics-Informed Neural Networks: A Second-Order Optimization Perspective.*
[3] Xu et al., 2025. *FP64 is All You Need: Rethinking Failure Modes in Physics-Informed Neural Networks.* NeurIPS 2025.

---

### Meta-Review · Area_Chair_caYg · 2026-01-06

**Summary:**

This paper proposes ResPINNs, reframing PINNs as residual flows to mitigate gradient shattering and flow mismatch, and the rebuttal/discussion substantially improved the submission via clearer framing and added evidence. However, despite these improvements, the discussion still leaves a material disagreement about whether the theory and evaluation are sufficiently convincing to support the central causal claims and generality, making this a borderline case that I lean weak reject.

**Reviewer Concerns:**

Several key concerns were at least partially addressed (notably, missing baselines / harder tasks / clarification of design choices).  Still, a main outstanding issue is that the theoretical story is not fully convincing as stated along with lingering questions about experimental design of evaluation.

**Reviewer Scores:**

The final signal is mixed: one reviewer is clearly positive (6tTh: 8/accept).  Another moved from moderate to strong accept (z8Qq: 6 → 8, explicitly supports acceptance). A marginal reviewer indicated an upward update (WL8d: 4/below threshold initially, later “increased my rating”). However, there remains a firm dissent (1kCn: 2/reject with confidence 5, centered on the persuasiveness of theory and strength of evaluation). Given this high-variance score profile and the unresolved high-confidence concerns, my overall recommendation is Weak Reject.

---

### Decision · Program_Chairs · 2026-01-26

Reject